# Serotonin reuptake inhibitors improve muscle stem cell function and muscle regeneration in male mice

Mylène Fefeu[1,2,3], Michael Blatzer[2], Anita Kneppers [4,11], David Briand [2,11], Pierre Rocheteau[2,11], Alexandre Haroche[1], David Hardy [2], Mélanie Juchet-Martin[2], Anne Danckaert [5], François Coudoré[6], Abdulkarim Tutakhail [6], Corinne Huchet[7], Aude Lafoux[8], Rémi Mounier[4], Olivier Mir[9], Raphaël Gaillard[1,2,3,12] ✉ & Fabrice Chrétien [2,3,10,12] ✉

Serotonin reuptake inhibitor antidepressants such as fluoxetine are widely used to treat mood disorders. The mechanisms of action include an increase in extracellular level of serotonin, neurogenesis, and growth of vessels in the brain. We investigated whether fluoxetine could have broader peripheral regenerative properties. Following prolonged administration of fluoxetine in male mice, we showed that fluoxetine increases the number of muscle stem cells and muscle angiogenesis, associated with positive changes in skeletal muscle function. Fluoxetine also improved skeletal muscle regeneration after single and multiples injuries with an increased muscle stem cells pool and vessel density associated with reduced fibrotic lesions and inflammation. Mice devoid of peripheral serotonin treated with fluoxetine did not exhibit beneficial effects during muscle regeneration. Specifically, pharmacological, and genetic inactivation of the 5-HT1B subtype serotonin receptor also abolished the enhanced regenerative process induced by fluoxetine. We highlight here a regenerative property of serotonin on skeletal muscle.

Since its approval four decades ago, fluoxetine (FLX), known as the first selective serotonin reuptake inhibitor (SSRI)[1], has become the world's most prescribed antidepressant[2]. Extensive research on neurotransmission, in particular the transmission of serotonin (also termed 5-hydroxytryptamine, 5-HT) in depressive disorders, has provided strong evidence on the mechanisms of action of FLX and other antidepressants, beyond the initial increase in extracellular serotonin levels that they induced[3,4]. Indeed, their antidepressant effects involve

an increase in hippocampal neurogenesis, neuronal survival[5–8] supported by a concomitant and mandatory increase in cerebral angiogenesis[9,10].

In addition to its well-known neurotransmission activity, serotonin is also described as a peripheral paracrine/autocrine hormone involved in various non-neuronal physiological functions[11]. As a potent mitogen, it can modulate tissue remodeling, and is released at the site of tissue injury by platelets[12,13]. Using *ad integrum* skeletal muscle

[1]GHU Paris Psychiatrie & Neurosciences, site Sainte Anne, Service Hospitalo-Universitaire de psychiatrie, Paris, France. [2]Institut Pasteur, Experimental Neuropathology Unit, Global Health Department, Paris, France. [3]Université de Paris Cité, Paris, France. [4]Institut NeuroMyoGène, Unité Physiopathologie et Génétique du Neurone et du Muscle, Université Claude Bernard Lyon 1, CNRS UMR 5261, Inserm U1315, Univ Lyon, Lyon, France. [5]Institut Pasteur, UTechS PBI, C2RT Paris, France. [6]CESP, MOODS Team, Inserm, Faculté de Pharmacie, Université Paris-Saclay, Châtenay-Malabry, France. [7]TaRGeT, INSERM UMR 1089, Nantes Université, CHU Nantes, Nantes, France. [8]Therassay Platform, Capacités, Université de Nantes, IRS 2 Nantes Biotech, Nantes, France. [9]Sarcoma Group, Gustave Roussy, Villejuif, France. [10]GHU Paris Psychiatrie & Neurosciences, site Sainte Anne, Service Hospitalo-Universitaire de neuropathologie, Paris, France. [11]These authors contributed equally: Anita Kneppers, David Briand, Pierre Rocheteau. [12]These authors jointly supervised this work: Raphaël Gaillard, Fabrice Chrétien. ✉e-mail: raphael.gaillard@normalesup.org; f.chretien@ghu-paris.fr

regeneration as a paradigm for regenerative medicine[14], we tested in mice whether FLX could have broader peripheral regenerative properties. We identified effects of FLX, accelerating muscle regeneration after single as well as multiple muscle injuries by increasing muscle stem cells (as known as satellite cells, SCs) without exhausting this stem cell pool. In addition, FLX enhanced muscle repair with a coordinated increase in vessel formation as well as a reduction in the inflammatory response. Moreover, we demonstrated that FLX effects were dependent of peripheral serotonin and that the 5-HT1B receptor was a major modulator of downstream FLX action in muscle stem cells.

## Results

### Fluoxetine increases muscle stem cells and vessels proliferation with improved muscle function

To investigate the impact of FLX on skeletal muscle stem cells, FLX was delivered *per os* at 18 mg/kg daily for six weeks to *Tg:Pax7-nGFP* mice[15], in which the GFP reporter gene marks all SCs (Fig. 1a). Histological (illustrated in Fig. 1b) and cytometric (illustrated in Supplementary Fig. 1a) analysis revealed that the SCs count in the *Tibialis anterior* (TA) increased by 60% to 80% with FLX treatment, respectively (Fig. 1c and Supplementary Fig. 1b). Using Pax7 immunolabelling, these results were also confirmed in the wild-type mouse strain C57Bl6 (Supplementary Fig. 1c). Extensive histological analyses of the skeletal muscle showed no morphological change neither in the fiber number nor their area (Supplementary Fig. 1d, e). To quantify proliferation of SCs, BrdU was delivered during prolonged FLX treatment for up to twelve weeks and showed that 90% of the SCs (GFP + /BrdU + ) proliferated by the sixth week of FLX treatment and this effect was stable and sustained until twelve weeks of FLX treatment (Supplementary Fig. 1f). A short BrdU pulse 12 h before death demonstrated that most of the SCs were dividing during the fourth and fifth week of FLX administration (Supplementary Fig. 1g). After 6 weeks of FLX treatment, 4% of the SCs pool were dividing (Pax7 + /Ki67+ positive cells), which was associated with an increase in *Cyclin D1* gene expression (Supplementary Fig. 1h–i). Of note, at steady state, the increase in the SCs pool was not associated with a change in protein expression of the differentiation regulatory factor myogenin (Supplementary Fig. 1j).

Muscle capillaries participate to muscle stem cell niches and angiogenesis is crucial to support SC survival and for an efficient and coordinated muscle regeneration (angiogenesis/myogenesis coupling)[14]. To investigate the effect of FLX on muscle vessels, we delivered FLX to *Flk1^{GFP/+}* mice[16] allowing the visualization of the entire capillary network due to the GFP expression by endothelial cells. After six weeks of treatment, FLX-treated mice exhibited a 44% increase in their vessels number in TA sections (Fig. 1d, e). To confirm this enhanced vascularization, we used CD31 as a marker for endothelial cells. C57Bl6 mice that received FLX treatment for 6 weeks also showed an increase in the vessels number in the TA and *Soleus* muscle sections (Supplementary Fig. 1k–l). To confirm angiogenic effects of FLX, we used an in vivo angiogenesis assay by subcutaneously grafting Matrigel® plugs containing myoblasts in C57Bl6 recipient mice[17]. Immunostaining of CD31 confirmed that vascularization increased 2.9-fold in the plugs of the FLX-treated mice compared to untreated controls (Fig. 1f, g). Neovessels in plugs appeared morphologically normal and were functional as assessed by the presence of a basal lamina and a lumen containing red blood cells (Supplementary Fig. 1m).

Associated with the effects induced locally by FLX treatment in skeletal muscle, we also observed changes in serum profiles of cytokines, chemokines and growth factors, with a marked increase in the circulating level of FGFb (Table 1).

Interestingly, a 6 weeks FLX treatment was also associated with positive functional and physiological changes in skeletal muscle. Assessed by an in vivo grip test, FLX-treated mice showed a 14% increase in forelimb muscle strength compared to controls (Fig. 1h).

In addition, the physical performance of FLX-treated mice tested by treadmill exercise was improved with an increased in maximal aerobic velocity, distance traveled and exercise endurance (Fig. 1i and Supplementary Fig. 1n–o). Moreover, mice treated by FLX exhibited a metabolic change in muscle fiber typing, with an enrichment of slow type I and fast type IIA and IIB fibers and a decrease in type IIX fiber content in the TA muscle (Fig. 1j and Supplementary Fig. 1p).

Taken together, these results suggest that chronic administration of FLX has a stable and sustained effect promoting proliferation of muscle stem cells and vessels associated with positive functional and metabolic changes.

### Fluoxetine accelerates muscle regeneration after injury in vivo

To investigate FLX´s regenerative capacity on skeletal muscles, we delivered FLX for six weeks and then performed notexin-induced injuries (phospholipase that induces a severe muscle necrosis) of the TA muscle in *Tg:Pax7-nGFP* and C57BL6 mice[18] (Fig. 2a). At four days post injury, FLX-treated mice showed a 2.5-fold increase in the number of SCs (Fig. 2b, c) with an increase in the levels of dividing cells (Supplementary Fig. 2a, b), and more cells were differentiating based on the differentiation regulatory factor myogenin immunolabelling (Fig. 2d, e, Supplementary Fig. 2c, d). Comparing muscles fourteen days post injury, we found a faster and more complete regeneration in FLX-treated mice marked by an increase in the fibers number probably newly formed due to their smaller size associated with a massive increase in regenerated fibers (i.e., 51% of centro-nucleated myofibers (CNF) in the FLX group *versus* 39% of centro-nucleated fibers in the control group) (Fig. 2f–h, Supplementary Fig. 2e, f). In addition, FLX-treated mice showed milder inflammatory infiltrates, minimal calcium deposits and lower endomysial collagen deposit fourteen days post NTX (Supplementary Fig. 2g–l). Twenty-eight days post injury the muscles were fully regenerated in both groups and appeared histologically similar to pre-injury (Supplementary Fig. 2m–o), except for centro-located nuclei in virtually all myofibers indicating regenerated fibers. Interestingly, the pool of SCs and the number of muscle capillaries were still significantly increased in FLX-treated animals after the complete regeneration of the muscle (Supplementary Fig. 2p–s).

To investigate the functional effects of FLX during muscle regeneration, we assessed in situ contractile parameters of TA muscle fourteen days post injury. FLX-treated mice exhibited an increase in the relative amplitudes of twitch and tetanos contractile responses by 22% and 27%, respectively (Fig. 2i, j).

To further examine the muscle regenerative capacity and SCs behavior, we next performed three repetitive muscle injuries (Fig. 2k) and found that the muscle efficiently regenerated without exhaustion of the regenerative capacity over several injuries and especially SCs counts (Fig. 2l, m).

Taken together, these results support that administration of FLX accelerates muscle regeneration upon injury by inducing beneficial effects on the three main components involved in this process via *(i)* an increase in the stem cells pool and their differentiation capacities without their exhaustion, associated with *(ii)* an increase in muscle vascularity as well as *(iii)* an overall decrease in the inflammatory response. Importantly, this improvement in muscle regeneration by FLX treatment is also associated with functional benefits on muscle contractile performance.

### Fluoxetine exerts cell autonomous on muscle stem cells by requiring stimulation of the 5-HT1B receptor

To investigate the effects of FLX directly on muscle stem cells, because the serum contained relatively high levels of serotonin (5-HT) (Supplementary Fig. 3a), we first studied in vitro the effects of 5-HT and FLX on immortalized murine myoblasts (C2C12) due to their cell viability in 5-HT-free medium (i.e, charcoal FBS). In addition, C2C12 cells

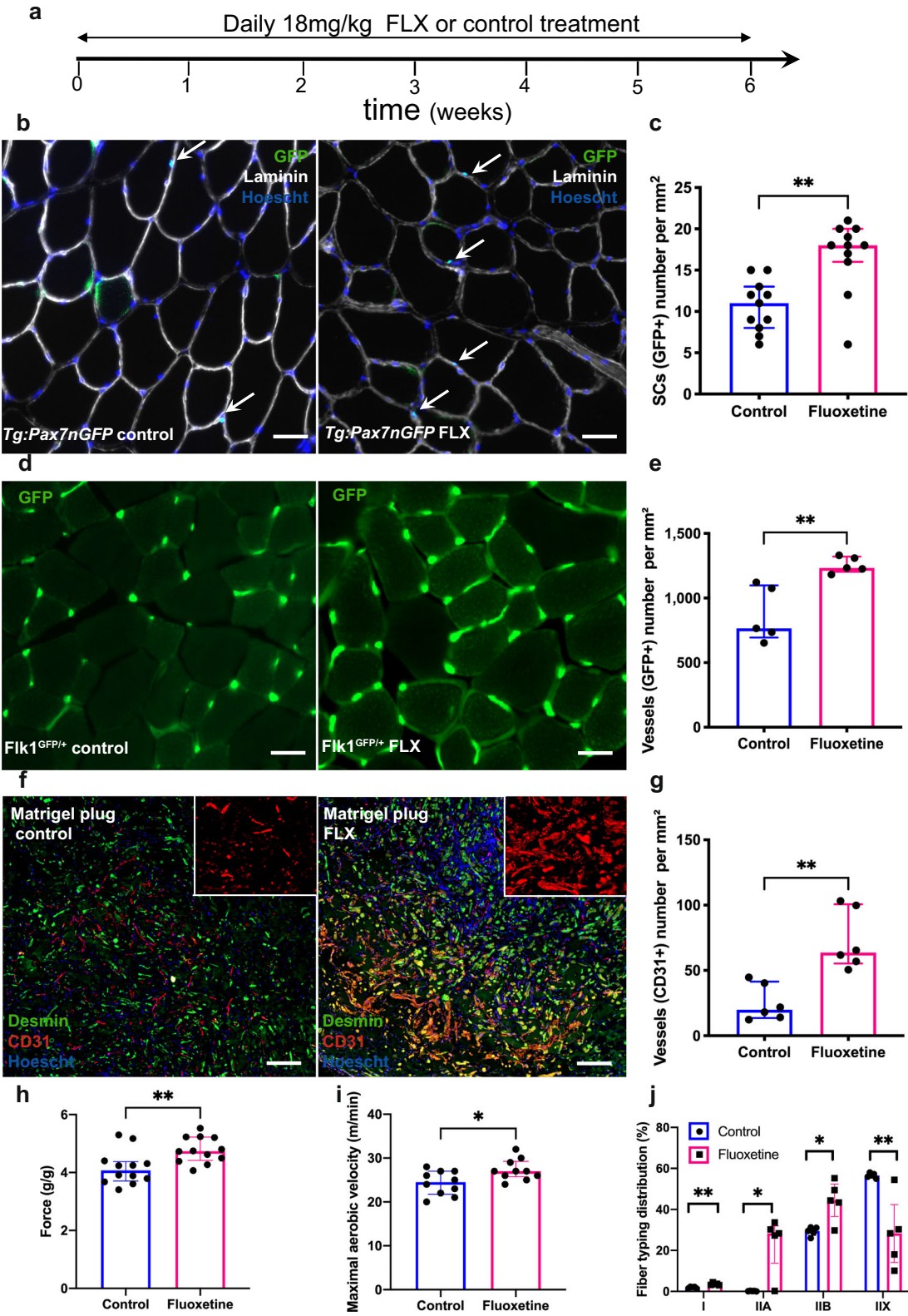

expressed SERT, the 5-HT transporter, known to be the primary target of FLX[1] (Supplementary Fig. 3b). Two days post plating in 5-HT-free medium, C2C12s exposed to exogenous 5-HT showed a 39.8% increase in relative nuclei number compared to controls (Supplementary Fig. 3c). Interestingly, C2C12s exposed only to FLX in 5-HT-free medium showed a similar number of nuclei as controls, whereas the nuclei number increased 1.5-fold after combined exposure to both of FLX and exogenous 5-HT (Supplementary Fig. 3c). In addition, the effects of

5-HT on C2C12 cell division rates showed a typical dose-response relationship (Supplementary Fig. 3d).

To identify the 5-HT's mode of action on skeletal muscle, we performed western blots for 5-HT receptors families 1 and 2 in TA muscle from mice treated for six weeks with FLX or control. Although 5-HT2A receptor expression has been previously described in skeletal muscle[19], here we determined a 108-fold upregulation of 5-HT1B receptor protein expression after FLX treatment (Fig. 3a) and

**Fig. 1 | Fluoxetine increases the proliferation of skeletal muscle stem cells, the functional vascularization, and the muscle strength of the skeletal muscle.**
**a** Schematic representation of fluoxetine (FLX) delivery. **b** Representative immunostaining TA sections of control and FLX-treated mice; sections display GFP (SCs marker, green), Laminin (extracellular matrix, white), Hoechst (nuclei, blue), the arrows indicate SCs. Scale bars indicate 50 μm. **c** Quantification of the SCs number by GFP immunostaining on TA sections from control and FLX-treated *Tg:Pax7nGFP* mice (n = 11 mice per condition from 2 independent experiments, p = 0.0014). **d** Representative immunostaining TA sections of control and FLX-treated *Flk1^GFP/+* mice; sections display GFP (endothelial cells marker, green). Scale bars indicate 50 μm. **e** Quantification of the vessels number by GFP immunostaining on TA sections from control and FLX-treated *Flk1^GFP/+* mice (n = 5 mice per condition, p = 0.0079). **f** Representative immunostaining Matrigel® plugs sections of control and FLX-treated C57Bl6 mice; sections display CD31+ cells (endothelial cell marker, red), Desmin (myoblasts marker, green), Hoechst (nuclei, blue). Scale bars indicate 50 μm. **g** Quantification of the vessels number by CD31 immunostaining on cross sections of Matrigel® plugs from control and FLX-treated recipient mice (n = 6 mice per condition, p = 0.0022). **h** In vivo forelimb grip strength of control and FLX-treated C57Bl6 mice (n = 12 mice per condition from 2 independent experiments, p = 0.0068). **i** Maximal aerobic velocity (VO₂ max in m/min) determined by treadmill exercise of FLX-treated and control C57Bl6 mice (n = 10 mice per condition from 2 independent experiments, p = 0.0391). **j** Quantification of percent fiber typing by MyHC subtypes immunostaining on TA sections from control and FLX-treated C57Bl6 mice (n = 5 mice per condition, p = 0.0079, p = 0.0159, p = 0.0317, p = 0.0079, respectively). All values are represented as median with interquartile range. All data analyses were performed with the two-tailed Mann–Whitney test. *p ≤ 0.05, **p ≤ 0.01. Source data are provided as a Source Data file.

confirmed this 5-HT1B receptor protein expression by C2C12s (Fig. 3b). To confirm the involvement of 5-HT1B receptor in the cellular effects of 5-HT and FLX, we investigated the number of C2C12 nuclei exposed to 5-HT and FLX after pretreatment with specific 5-HT1B antagonists (i.e., SB216641 and GR127935). The increase in the number of C2C12 nuclei following exposure to 5-HT or to FLX was abolished with both 5-HT1B

### Table 1 | Circulating levels of cytokines, chemokines, and growth factors after prolonged fluoxetine treatment at steady state

| Cytokines levels | Control (n = 9) | Fluoxetine (n = 9) | | |
|---|---|---|---|---|
| (pg/ml) | Mean ± SEM | Mean ± SEM | % of change | p |
| IL-1α | 41 ± 8 | 35 ± 14 | –15% | NS |
| IL-1β | 220 ± 66 | 217 ± 92 | –1% | NS |
| IL-2 | 0 ± 0 | 0 ± 0 | – | – |
| IL-3 | 23 ± 5 | 30 ± 14 | 32% | NS |
| IL-4 | 0 ± 0 | 0 ± 0 | – | – |
| IL-5 | 0 ± 0 | 0 ± 0 | – | – |
| IL-6 | 0 ± 0 | 0 ± 0 | – | – |
| IL-9 | 0 ± 0 | 0 ± 0 | – | – |
| IL-10 | 29 ± 8 | 33 ± 13 | 13% | NS |
| IL-12(p40) | 97 ± 18 | 126 ± 21 | 29% | 0.019 |
| IL-12(p70) | 479 ± 140 | 469 ± 253 | –2% | NS |
| IL-13 | 400 ± 111 | 622 ± 166 | 55% | 0.012 |
| IL-17 | 0 ± 0 | 0 ± 0 | – | – |
| Eotaxin | 865 ± 220 | 1131 ± 490 | 30% | NS |
| G-CSF | 59 ± 20 | 57 ± 24 | –4% | NS |
| GM-CSF | 145 ± 45 | 152 ± 30 | 4% | NS |
| IFN-γ | 13 ± 2 | 14 ± 3 | 11% | NS |
| KC | 0 ± 0 | 0 ± 0 | – | – |
| MCP-1 | 0 ± 0 | 0 ± 0 | – | – |
| MIP-1a | 80 ± 21 | 90 ± 21 | 12% | NS |
| MIP-1b | 19 ± 5 | 21 ± 9 | 6% | NS |
| RANTES | 37 ± 14 | 26 ± 15 | –27% | NS |
| TNF-α | 1983 ± 578 | 2095 ± 736 | 6% | NS |
| FGF-b | 70 ± 47 | 438 ± 123 | 421% | 0.0087 |
| MIG | 368 ± 151 | 400 ± 154 | 8% | NS |
| PDGF-β | 1503 ± 975 | 1304 ± 1069 | –13% | NS |
| VEGF | 16 ± 2 | 17 ± 8 | 8% | NS |

Multiplex Elisa (Luminex® assay) was performed on the serum of control and 6 weeks fluoxetine (FLX)-treated mice (n = 9 per condition). The values are represented as the mean ± standard error of mean (SEM) and the percentage of fold-change relative to control mice were also calculated. In red are represented the percentage of fold-change increases compared to the control group and in blue are represented the percentage of fold-change decreases compared to the control group. Individual p values were determined with the two-tailed Mann–Whitney test.

antagonists (Supplementary Fig. 3e, f) while it was maintained with a specific 5-HT2A antagonist (i.e., MDL100907) as a control (Supplementary Fig. 3f). To identify in depth the signaling pathways involved by 5-HT1B receptor stimulation, we investigated the levels of cAMP, known as the major second messenger of this receptor[20] and screened the relative levels of phosphorylation of major protein kinases in C2C12s. We showed that short exposure to 5-HT or FLX led to a decrease in intracellular cAMP level, an effect abolished with 5-HT1B antagonist (Supplementary Fig. 3g, h). After a short stimulation with 5-HT, the most phosphorylated kinases were ERK2, PYK2 or p27 and the least phosphorylated kinases were WNK1, Fgr or p53. These 5-HT-induced changes were also counteracted when C2C12s were previously exposed to a 5-HT1B antagonist (Supplementary Fig. 3i).

Finally, we showed by RT-qPCR and western blot the specific expression of 5-HT1B receptor in SCs (Fig. 3b and Supplementary Fig. 3j) and we also confirmed the in vitro effects of FLX on FACS-sorted SCs from *Tg:Pax7nGFP* mice. Indeed, SCs exposed to FLX over time showed an increase in the relative number of nuclei compared to controls (Supplementary Fig. 3k). At four days post plating, SCs exposed to FLX showed an increased percentage of myoblasts expressing Pax7/GFP and differentiating cells expressing myogenin (Fig. 3c, d). At fourteen days post plating, the FLX condition was associated with an increase in the fusion index, a marker of late stage of myoblast differentiation and an increase in the number of SCs expressing Pax7/GFP (Fig.3e, f). These effects were abolished by a 5-HT1B antagonist (Fig. 3c–f and Supplementary Fig. 3k). Ex vivo, we also showed that exposure to 5-HT for four days promoted myogenin expression by SCs cultivated from single fibers (Fig. 3g, h). Using culture of FACS-sorted SCs with serum from FLX-treated or control mice, we observed by videomicroscopy that the first division of SCs occurred faster, and the division rate was higher with FLX serum (Fig. 3i, j). Similar to the FLX effects described previously, SCs cultured in FLX serum displayed an increased number of myogenin+ cells at four days post plating and Pax7+ cells at fourteen days post plating (Fig. 3k, l). These effects were also abolished by a 5-HT1B antagonist (Fig. 3i–l).

Taken together, these in vitro results suggest that fluoxetine exerts autonomous effects on SCs by promoting activation with entry into the cell cycle, proliferation, early and terminal differentiation and, finally, the self-renewal of SCs. Moreover, FLX acts in a serotonin-dependent manner on SCs and more specifically via the 5-HT1B receptor, suggesting that FLX, by blocking SERT, induces an increase in the level of extracellular 5-HT leading to an increased stimulation of the 5-HT1B receptor.

### The beneficial effects of fluoxetine on muscle regeneration in vivo are also mediated by 5-HT1B receptor

To demonstrate the 5-HT-dependent effects of FLX on SCs in vivo, we delivered FLX to Tryptophan Hydroxylase 1 knockout mice (TPH1^−/−, which lack serotonin in peripheral organs[21]. FLX-treated *TPH1^−/−* mice did not exhibit an increase in Pax7 + -SCs number on uninjured TA

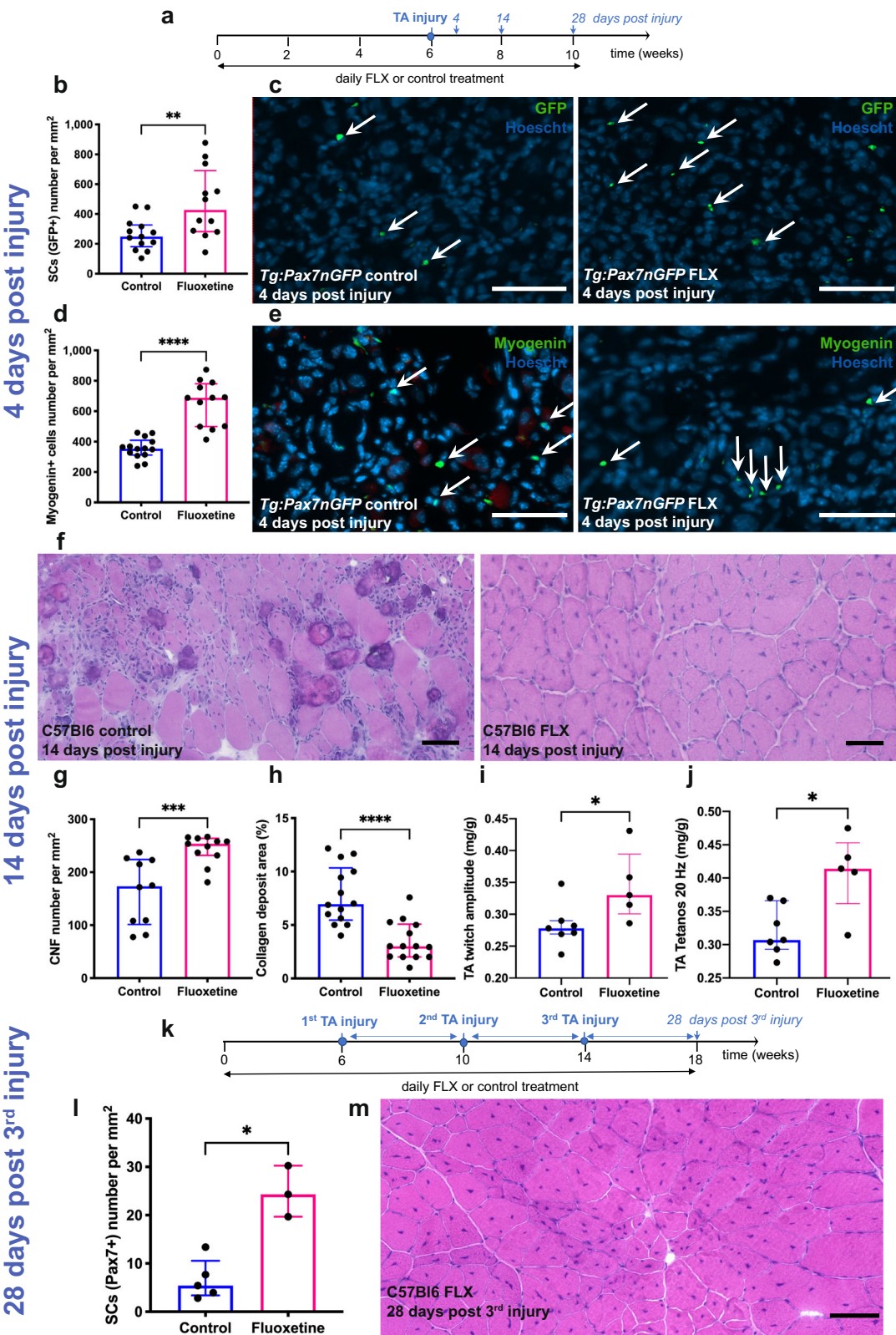

sections, unlike FLX-treated wild-type C57Bl6 mice (Supplementary Fig. 4a). Of note, the untreated *TPH1*[−/−] mice didn't show significant change in the SCs number compared to the wild-type mice (Supplementary Fig. 4a). At four days post injury, FLX-treated *TPH1*[−/−] mice showed neither an increase in Pax7[+] SCs (Fig. 4a) nor an increase in myogenin[+] differentiating cells (Fig. 4b) compared to FLX-treated wild-type mice. In addition, in these FLX-treated *TPH1*[−/−] mice, no reduction

of collagen deposits areas was observed fourteen days post injury (Supplementary Fig. 4b).

To investigate whether FLX's effects on muscle stem cells and vessels are mediated by 5-HT1B receptor in wild-type C56Bl6 mice we co-administered a selective 5-HT1B antagonist (GR127935) throughout the six weeks of FLX treatment. In uninjured muscles, mice treated with FLX and GR127935 did not show an increase in the number of SCs

**Fig. 2 | Muscle regeneration after injuries is effectively improved by fluoxetine.** **a** Schematic representation of the notexin-induced TA injury model, fluoxetine (FLX) administration and death time points. **b** Quantification of SCs number by GFP immunostaining 4 days post injury on TA sections from control and FLX-treated *Tg:Pax7nGFP* mice (*n* = 12 FLX and 13 control, 2 independent experiments, *p* = 0.0096). **c** Representative immunostaining TA sections of control and FLX-treated mice 4 days post injury; sections display GFP (green), Laminin (white), Hoechst (nuclei, blue), the arrows indicate SCs. **d** Quantification of differentiating cells number by Myogenin immunostaining 4 days post injury on TA sections from control and FLX-treated *Tg:Pax7nGFP* mice (*n* = 12 per condition, 2 independent experiments, *p* < 0.0001). **e** Representative immunostaining TA sections of control and FLX-treated mice 4 days post injury; sections display Myogenin (green), Laminin (white), Hoechst (nuclei, blue), the arrows indicate SCs. Scale bars indicate 50 μm. **f** Representative Hematoxylin and Eosin (HE)-stained TA sections of control and FLX-treated mice 14 days post injury. **g** Quantification of the centro-nucleated fibers (CNF) number 14 days post injury on TA sections from control and FLX-

treated C57Bl6 mice (*n* = 10 control and 11 FLX, 2 independent experiments, *p* = 0.0008). **h** Percentage of the collagen deposit area stained with Sirius Red 14 days post injury on TA sections from control and FLX-treated *Tg:Pax7nGFP* mice (*n* = 14 per condition, 2 independent experiments, *p* < 0.0001). Relative amplitudes of in situ twitch **i** and tetanos at 20 Hz. **j** Contractile responses of TA from control and FLX-treated C57Bl6 mice (*n* = 5 FLX and 7 control, *p* = 0.0303 and *p* = 0.0177, respectively). **k** Schematic representation of the three repetitive notexin-induced TA injury model, FLX administration and death time points. **l** Quantification of the SCs number by Pax7 immunostaining 28 days post the third notexin injury on TA sections from control and FLX-treated C57Bl6 mice (*n* = 3 FLX and 5 control, *p* = 0.0357). **m** Representative HE-stained TA sections of FLX-treated mice 28 days post the third notexin injury. All values are represented as median with interquartile range. All scale bars indicate 50 μm. All data analyses were performed with the two-tailed Mann–Whitney test. *$p \leq 0.05$, **$p \leq 0.01$, ***$p \leq 0.001$, ****$p \leq 0.0001$. Source data are provided as a Source Data file.

nor an increase in the number of vessels, unlike the FLX-treated mice (Supplementary Fig. 4c, d).

To next determine the involvement of the 5-HT1B receptor in mediating FLX's effects on muscle regeneration, we administered the 5-HT1B antagonist GR127935 after six weeks of FLX treatment and before the notexin-induced muscle injury (Fig. 4c). At four days post injury, both increases in the number of SCs and differentiating cells showed by FLX-treated mice were not observed in mice receiving FLX and GR127935 (Fig. 4d, e). In addition, fourteen days post injury, the mice receiving FLX and GR127935 did not exhibit the decrease in collagen deposits area, the decrease in calcium deposit number and the decrease in immune (Gr1+ and F4/80 + ) cells infiltration observed in the FLX-treated mice (Supplementary Fig. 4e–h). As a specificity control, no abrogation of the FLX effects on muscle regeneration was observed when mice were treated with a selective 5-HT2A antagonist (MDL100907) and FLX (Supplementary Fig. 5a–g).

Finally, to unequivocally confirm the role of 5-HT1B receptor in FLX effects, we used floxed tetracycline operator–5HT1B (*tet*O1B) mice[22] crossed with *Pax7Cre*[ER(T2)] mice[23] to specifically and conditionally eliminate 5-HT1B receptor from SCs (Supplementary Fig. 4i). *Pax7-Cre*[ER(T2)]*::tetO1B* mice were treated with FLX and tamoxifen (Fig. 4f). C57Bl/6 with tamoxifen and *Pax7-Cre*[ER(T2)]*::tetO1B* without tamoxifen served as controls. In non-injured mice, the *Pax7-Cre*[ER(T2)]*::tetO1B* mice treated by tamoxifen and FLX showed a significant decrease in SCs number compared to the wild-type mice treated by tamoxifen and FLX and the *Pax7-Cre*[ER(T2)]*::tetO1B* mice treated by FLX (Supplementary Fig. 4j). At four days post injury, we observed no increase in the number of myogenin+ cells in the tamoxifen and FLX-treated *Pax7-Cre*[ER(T2)]*::tetO1B* (Fig. 4g). Collagen deposit area were also unchanged fourteen days post injury in tamoxifen + FLX-treated *Pax7-Cre*[ER(T2)]*::tetO1B* mice (Fig. 4h).

In summary, these results demonstrate that the positive effects on muscle regeneration induced by FLX treatment are 5HT-dependent and are specifically mediated via the 5-HT1B receptor expressed on the SCs.

## Discussion

Regenerative medicine is a broad field of translational research including, in particular, activator molecules which can favorably influence tissue self-repair processes[24]. Far from its original anti-depressant effects[1], we provide here a characterization of beneficial actions of fluoxetine (FLX) on mouse muscle regeneration which implement an impact on both muscle stem cells, vascularization and inflammation leading to a faster muscle healing with limited fibrosis. Importantly, this improvement muscle regeneration is also associated with functional benefits of skeletal muscle. Remarkably, this study establishes that satellite cells (SCs) are sensitive to the mitogenic potential of peripheral serotonin (5-HT) and that FLX, as a modulator

of bioavailable serotonin levels[1], positively influences the behavior of SCs via the 5-HT1B receptor.

We demonstrate by multiple and complementary experimental approaches that FLX promotes over time a controlled and sustained proliferation of SCs at steady state. Interestingly, using a classical model of muscle injury, we prove that FLX induces a faster, more efficient, and still harmonious muscle regeneration. At the early regeneration stages, FLX increases the number of Pax7+ and myogenin + cells which are myogenic markers of activation/proliferation and differentiation, respectively[25] and, at later stages, FLX leads to a more rapid increase in fiber number and more specifically of centro-nucleated fibers, thus fully regenerated. More importantly, after multiple muscle injuries, FLX preserves the regenerative capacities of the muscle with a persistently increased SC pool. These results show that all the properties of SCs, namely self-renewal, proliferation, and differentiation, are improved and remained functional during FLX treatment. To our best knowledge, this is the first time that the beneficial effects of the FDA-approved drug FLX on the behavior of muscle stem cells have been demonstrated.

Taking advantage of complementary in vitro and in vivo approaches, we demonstrate that FLX impacts primary SCs or immortalized myoblasts by extracellular or peripheral 5-HT levels. In vitro we have characterized this classic biphasic dose-response aspect of the pro-liferative effects of 5-HT previously described in other cell types[26–28]. Our results fully support the hypothesis of a serotoninergic identity of skeletal muscle in line with previous studies showing the expression of 5-HT system-related actors in myoblasts or muscle such as the synthetic TPH1 enzyme[29], the degradation monoamine oxidase enzyme[30] or the SERT transporter[31] associated with a 5-HT signal promoting myoblasts differentiation[29,32]. Using the murine genetic model TPH1[−/−] model constitutively devoid of peripheral 5-HT, we illustrate that the pool of SCs at steady state as well as their regenerative capacities are preserved in the absence of peripheral 5-HT, whether systemic or within a serotoninergic system, suggesting that this 5-HT booster signal is redundant for the development and maintenance of SCs. Furthermore, the abolition of the pro-proliferative and pro-differentiating effects of FLX on SCs post injury in TPH1[−/−] mice confirm the crucial role of 5-HT in FLX's mode of action. In addition, the lack of FLX effects on muscle regeneration in TPH1[−/−] mice precludes the hypothesis of an involvement of the typical actions of FLX targeting the central nervous system inducing behavioral changes in mice including locomotor activity[33]. Previously, using similar experimental approaches, the role of serotonin, independent of its central effects, has been demonstrated in the regulation of energy metabolism or the bone density turnover[34,35]. Interestingly, the involvement of peripheral 5-HT within serotoninergic microsystems in the positive or negative regulation of stem cell behavior[35–42] and more generally in tissue regeneration[12,13,43–45]

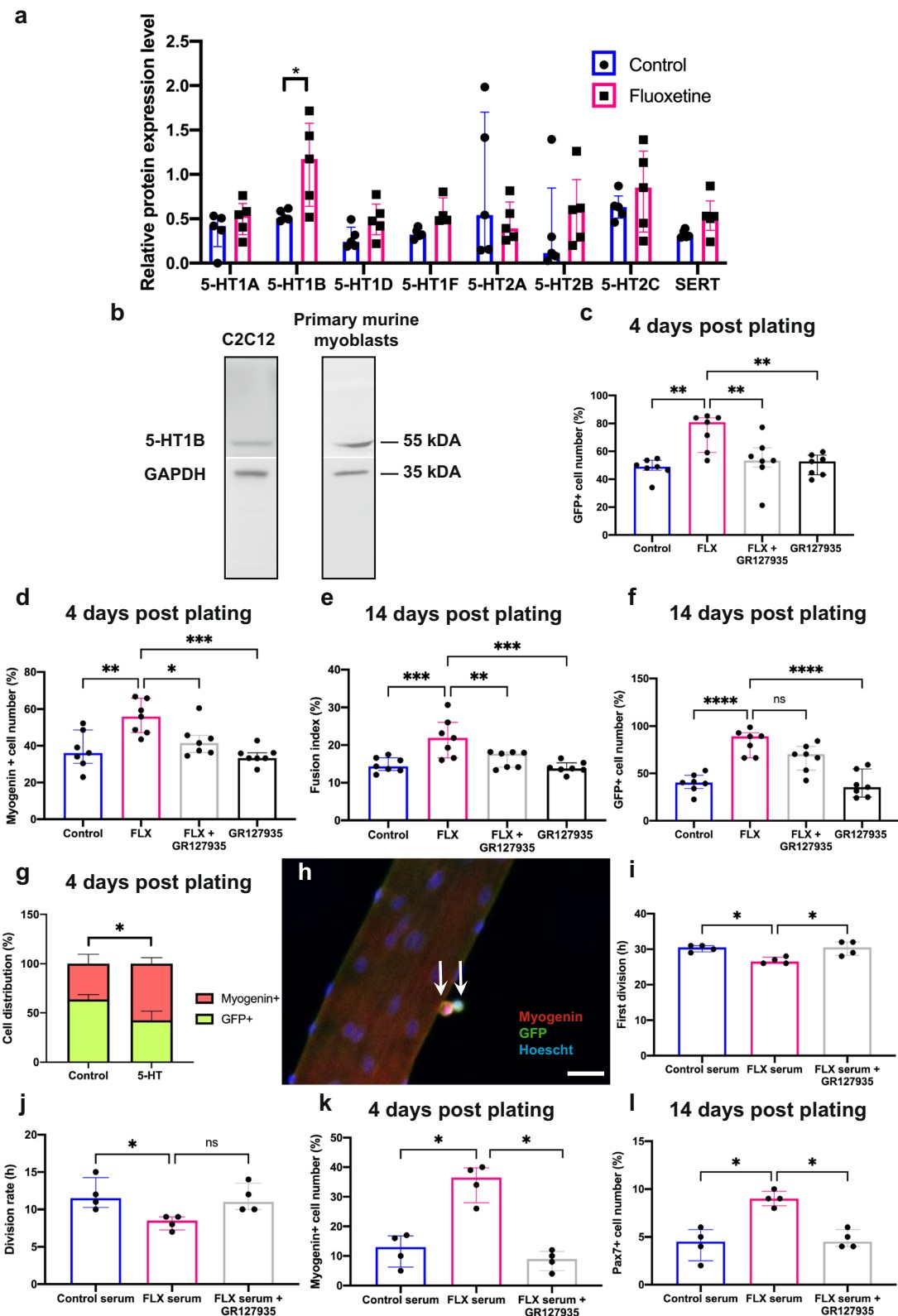

has been demonstrated in different organs, highlighting the high therapeutic potential of its pharmacological modulation.

Remarkably, pharmacological inactivation or genetic conditional inactivation models have shown that the FLX effects require stimulation of the 5-HT1B receptor specifically expressed in SCs. We propose as a mechanistic hypothesis underlying the cellular effects of FLX that it, by blocking the SERT transporter, increases the bioavailability of extracellular 5-HT allowing the reinforcement of the signal induced by

the activation of the 5-HT1B receptor leading to the decrease of intracellular levels of cAMP and to the positive or negative modulation of different key protein kinases in the functioning of SCs. However, further investigations are needed to decipher the downstream pathways regulated by 5-HT1B in SCs. Previous studies have reported an increasing gradient of 5-HT2A expression in myoblasts undergoing differentiation into myotubes with a metabolic role in glucose transport[19]. However, in our study, the in vivo use of a 5-HT2A

**Fig. 3 | The positive effects of fluoxetine on the stages of in vitro myogenesis are serotonin dependent and act through the 5-HT1B receptor. a** Relative level of protein expressions of various 5-HT receptors in TA muscle from control and fluoxetine (FLX)-treated C57Bl6 mice ($n = 5$ per condition, $p = 0.0317$).
**b** Representative western blot of the expressions of the targeted 5-HT1B receptor protein and the housekeeping GAPDH protein in C2C12 and murine primary myoblasts isolated by pre-platting from control mice ($n = 3$). Quantification of the relative number of (**c**) GFP+ (%) and (**d**) myogenin+ (%) among SCs exposed to FLX, FLX + GR127935 or GR127935 (5-HT1B antagonist) for 4 days ($n = 7$ per condition, $p = 0.0011$, $p = 0.0085$, $p = 0.0027$ for (**d**); $p = 0.0019$, $p = 0.0294$, $p = 0.0004$ for (**e**), respectively). Quantification of (**e**) the fusion index (%) and (**f**) the relative number of GFP+ (%) among SCs exposed to FLX, FLX + GR127935 or GR127935 for 14 days ($n = 7$ per condition, $p = 0.0007$, $p = 0.0058$, $p = 0.0002$ for (**f**), $p < 0.0001$ for (**g**), respectively). **g** Cell distribution (%) of GFP+ and myogenin+ among SCs from single fibers of *Tg:Pax7nGFP* mice exposed to 5-HT for 4 days ($n = 4$ per condition,

$p = 0.0473$). **h** Representative immunostaining of SCs from single fiber at 4 days post plating; sections display GFP (green), Myogenin (red), Hoechst (nuclei, blue), the arrows indicate SCs. Scale bars indicate 20 μm. **i** First cell division and (**j**) cell division rate assessed by live videomicroscopy from SCs plated for 5 days with control serum, FLX serum or FLX serum+GR127935 ($n = 4$ per condition, $p = 0.0458$, $p = 0.0402$ for (**j**); $p = 0.0230$ for (**k**), respectively). **k** Quantification of the relative number of myogenin+ (%) among SCs plated for 4 days with control serum, FLX serum or FLX serum + GR127935 ($n = 4$ per condition, $p = 0.0881$, $p = 0.0138$). **l** Quantification of the relative number of Pax7+ (%) among SCs plated for 14 days with control serum, FLX serum or FLX serum+GR127935 ($n = 4$ per condition, $p = 0.0261$, $p = 0.0448$). For figures (**c**)–(**f**) and (**i**)–(**l**), SCs were sorted by FACS from *Tg:Pax7nGFP* mice. All values are represented as median with interquartile range. The two-tailed Mann–Whitney test for (**a**), the two-tailed Fisher's test for (**g**) and the Kruskal-Wallis test for (**c**)–(**f**), (**i**)–(**l**). *$p \leq 0.05$, **$p \leq 0.01$, ***$p \leq 0.001$, ****$p \leq 0.0001$. Source data are provided as a Source Data file.

antagonist did not counteract the beneficial effects of FLX during muscle regeneration ruling out its involvement. The mediation of the 5-HT signal through these numerous receptors is complex[46]. Thus, several 5-HT receptors can be co-expressed within the same cell type in order to regulate distinct functions[47] and a sequential expression of different receptors specific to different stages of the stem cell maturation cascade has been reported in the brain or the bone marrow[38,48,49].

In other tissues, FLX similarly promotes the proliferation of other progenitor cells after prolonged administration[5–8,38]. In our study, we show an onset of response in the muscle marked by a significant increase in BrdU+ SCs from the fourth week, whereas the response of erythropoietic or neuronal progenitors were reported earlier[5–8,38]. The cell-type specific response to the proliferative effects of FLX could be explained by the sensitivity of cells to 5-HT signal as well as by the frequency of stem cells turnover. On one hand, SCs display a low turnover rate compared to other progenitor cells[50]. On the other hand, contrary to the brain[51] or the bone marrow[37], our results suggest that, the 5-HT signal is redundant in the functioning of SCs. Furthermore, we show a plateau effect of FLX on the SCs pool saturating at 90% of cumulative BrdU+ cells during prolonged treatment, suggesting that 10% SCs seem insensitive to the FLX effects. The population of SCs is known to be heterogenous, in particular with committed myogenic progenitors and around of reserve stem cells[25,52]. Interestingly, FLX positively impacts early neuronal progenitors with no effect on neural stem cells[5]. Further studies are needed to clarify whether the FLX effects differ depending on the type of SCs.

Associated with these specific effects targeting SCs, FLX induces a modification of the microenvironment of SCs with a coordinated increase in vascularization and in circulating levels of growth factors such as FGFb. The increased angiogenic capacities of FLX were confirmed by an in situ angiogenesis assay. Thus, FLX therefore enhances a known bidirectional interaction promoting their respective proliferation between endothelial cells and SCs[53]. In addition, the beneficial effects of FLX on the vascular niche reinforces the virtuous loop induced by endothelial cells in the maintenance and functioning of SCs[54]. Similarly in the brain, the FLX effects on neurogenesis also involved mandatory crosstalk with endothelial cells[9,55,56]. Moreover, the abolition of the FLX proangiogenic effects by a 5-HT1B antagonist in muscle suggests that these effects are mediated by 5-HT. The 5-HT involvement in angiogenesis has been extensively studied showing its role in the proliferation and migration of endothelial cells or even neovascularization by sprouting[27,57] with participation in particular of 5-HT1B receptor expressed by endothelial cells[57,58].

In support FLX also shows action on major players in muscle regeneration, namely the immune system and connective tissue[59]. Indeed, concomitant with coordinated and accelerated regeneration, a decrease in immune cell infiltration and collagen deposits are

observed in the muscle of mice treated with FLX, witnessing efficient tissue healing. In addition, the FLX-induced decrease in immune cells infiltration including macrophages was specifically abolished by a systemic 5-HT1B antagonist. The role of macrophages in particular their balance between pro-inflammatory M1 and pro-regenerative M2 phenotypes, is crucial during muscle regeneration[59]. Interestingly, among the many 5-HT mediated immunomodulatory effects[60], 5-HT has been reported to modulate the polarization of macrophages promoting maintenance of M2 status[61]. However, these effects were mediated by 5-HT2B and 5-HT7 receptors expressed by macrophages and further studies would be needed to clarify the role of 5-HT1B receptor in the FLX effects on macrophages. Finally, in our study, the antifibrotic effects of FLX were counteracted in the absence of peripheral 5-HT or 5-HT1B activation in SCs and post injury collagen deposition was similar between control or TPH1−/− groups. These results contrast with previous studies showing the pro-fibrotic role of 5-HT mediated by 5-HT2B and TGFβ expressed by fibroblasts and phenotypic improvement in TPH1−/− in a model of skin fibrosis[62]. Further studies are therefore needed to clarify the interactions between SCs expressing 5-HT1B receptor and fibroblasts expressing 5-HT2B receptor in the regulation of fibrotic processes usually considered as the substratum of symptoms in muscle diseases such as muscular dystrophies.

The combined action of FLX on myogenesis, angiogenesis and inflammation supporting harmonious muscle regeneration in an acute injury model associated with functional benefits, makes it a worthy therapeutic strategy for endogenous muscle diseases. Interestingly, FLX has already been shown to have a major positive impact on muscle strength, possibly dependent on peripheral serotonin[63,64] and several studies point to the potential effects of FLX in a model of chronic muscle injury such as Duchenne muscular dystrophy[65–67]. Beyond muscle our results point out that the effects of antidepressants on the regeneration of other organs than the brain need to be investigated. As such, our work opens new perspectives to further study the role of serotonin and 5-HT1B receptor on other stem cell types and to further understand the role of antidepressants on stem cell behavior and their therapeutic potential on regenerative medicine.

## Methods
### Mice and animal procedures
All procedures in this study were approved by the Animal Care and Use committee at the Institut Pasteur (CETEA 2014-0040). For the in situ experiments, all procedures were conducted in conformity with European rules for animal experimentation (French Ethical Committee APAFIS#33392-2021100416152925, November 19, 2021). The report on animal experiments complied with the ARRIVE guidelines. Unless specified 6 weeks old male mice were used in this study and different strains were used: the wild-type C57Bl/6RJ, *Tg:Pax7nGFP*[15], *Flk1*[GFP/+ 16], TPH1[−/−21], *Pax7-Cre*[ER(T2)23] and *tetO1B*[22]. All the

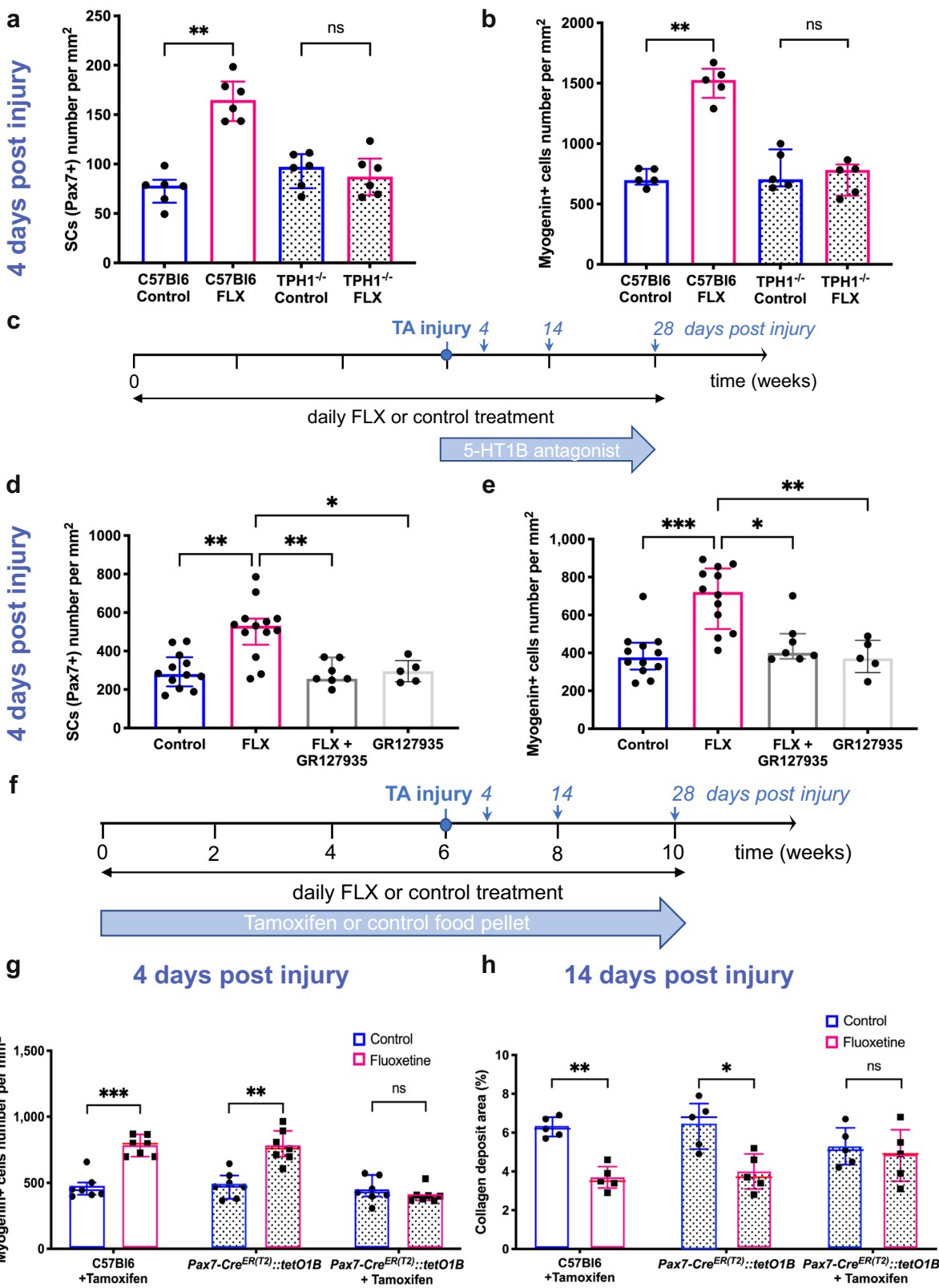

genotyping were performed by standard PCR methods. Mice were housed on a 12:12 light/dark cycle in a pathogen free facility with controlled temperature and humidity. Food and drink were given *ad libitum*. All experiments included a control group, and there were no exclusion criteria during the study. All animals were simultaneously randomized to the different control or treatment groups without considering any other variable. The animals were euthanized by cervical dislocation, as approved by the Animal Care and Use committee at the Institut Pasteur.

Fluoxetine (Lilly, fluoxetine hydrochloride syrup 20 mg/5 mL or Arrow, chlorhydrate fluoxetine syrup 20 mg/5 mL or Anawa Trading, fluoxetine hydrochloride) was dissolved in the drinking water and delivered *per os* at 18 mg/Kg. Antagonists GR127935 (Potent and selective 5-HT1B/1D antagonist, Tocris #1477) and MDL100907

**Fig. 4 | The beneficial effects of fluoxetine are also mediated in vivo by 5-HT1B receptor during muscle regeneration. a** Quantification of the SCs number by Pax7 immunostaining 4 days post injury on TA sections from control and fluoxetine (FLX)-treated C57Bl6 and *TPH1⁻/⁻* mice (*n* = 6 mice per condition, *p* = 0.0022). **b** Quantification of the differentiating cells number by Myogenin immunostaining 4 days post injury on TA sections from control and FLX-treated C57Bl6 and *TPH1⁻/⁻* mice (*n* = 5 mice per condition, *p* = 0.0079). **c** Schematic representation of the notexin-induced TA injury model, FLX and 5-HT1B antagonist (GR127935) administrations and death time points. **d** Quantification of the SCs number by Pax7 immunostaining 4 days post injury on TA sections from control, FLX and FLX + GR127935 and GR127935-treated C57Bl6 mice (*n* = 13 FLX, 12 control, 7 GR127935, 5 GR127935; 2 independent experiments, *p* = 0.0025, *p* = 0.0064, *p* = 0.0347, respectively). **e** Quantification of the differentiating cells number by Myogenin immunostaining 4 days post injury on TA sections from control, FLX, FLX + GR127935 and GR127935-treated C57Bl6 mice (*n* = 12 FLX, 12 control, 7 FLX + GR127935, 5 GR127935; 2 independent experiments, *p* = 0.0002, *p* = 0.0439, *p* = 0.0079, respectively). **f** Schematic representation of the notexin-induced TA injury model, FLX and tamoxifen administrations and death time points. **g** Quantification of the differentiating cells number by Myogenin immunostaining 4 days post injury on TA sections from C57Bl6 and *Pax7-Creᴱᴿ⁽ᵀ²⁾::tetO1B* mice receiving the control, tamoxifen and/or FLX (*n* = 7 mice per condition, *p* = 0.0022, *p* = 0.0012, respectively). **h** Percentage of the collagen deposit area stained with Sirius Red 14 days post injury on TA sections from C57Bl6 and *Pax7-Creᴱᴿ⁽ᵀ²⁾::tetO1B* mice receiving the control, tamoxifen and/or FLX (*n* = 5 mice per condition, *p* = 0.0079, *p* = 0.0159, respectively). All values are represented as median with interquartile range. The two-tailed Mann–Whitney test for (**a**), (**b**), (**g**) and the Kruskal-Wallis test for (**d**), (**e**), (**h**). *\*p* ≤ 0.05, *\*\*p* ≤ 0.01, *\*\*\*p* ≤ 0.001. Source data are provided as a Source Data file.

---

(Potent and selective 5-HT2A antagonist, Tocris #4173) were used at 4 mg/Kg respectively and delivered in Azlet osmotic sub-cutaneous pump (#2001D).

Thymidine analog Bromodesoxyuridine (BrdU, Sigma-Aldrich #B5002) was used to assess cell proliferation at a concentration of 50 mg/kg dissolved in 0.9% NaCl. For the short pulse chase labeling, the mice received intraperitoneally BrdU 12 h and 4 h before death. For the cumulative assay BrdU was delivered *per os* in the drinking water during 12 weeks throughout the course of fluoxetine or control condition.

For muscle injuries, mice were anesthetized with isoflurane, the back limb was shaved and injected with a Hamilton syringe containing 10 μl of notexin (LATOXAN #L8104) at 0.25 mg/ml directly in the *Tibialis anterior* (TA). We have chosen 4 days post-injury as it corresponds to a significant expression of Myogenin; 14 days as it corresponds to half regenerated muscle in normal conditions and 28 days for fully regenerated muscle. In the serial injury experiments, notexin was injected every 28 days and a subgroup of mice was sacrificed to control for efficient regeneration.

To induce the Cre-mediated recombination, the *Pax7-Creᴱᴿ⁽ᵀ²⁾::tetO1B* mice were given Tamoxifen food pellets (TAMOXIFEN DIET TAM400/CreER, Harlan TD.55125) for 21 days or throughout FLX administration.

### In vivo force measurement
A grip strength meter (Ugo Basile ®, Italy), equipped with DCA software version and attached to a force transducer, measured the gripping strength of forelimb. Each mouse was placed over a base plate in front of a grid. Five trials were conducted for each mouse alternatively. Results are expressed as the result of tree peak forces (in g), normalized to the body weight (in g). The experimental procedures were repeated independently two times.

### Treadmill exercise
A 6-lane motorized treadmill for rodents (Ugo Basile®, Italy) equipped with UB X-Pad version 1.0.01 software that automatically recorded the animal's distance, speed and running time was used for exercise. To reduce their stress, the mice were habituated to treadmill exercise for one week of adaptation (20 min/day, 15 m/min). To reach the individual maximum aerobic running speed, the treadmill belt speed was increased by 0.03 m/s every 2 min from 6 m/min until exhaustion. The treadmill was used to determine the physical performance of each mouse: maximum aerobic velocity (VO₂ max in m/min), time to exhaustion in VO₂ max 75% (VO₂ max ₇₅% = VO₂ max × 0.75) and running distance in VO₂ max 75%. The experimental procedures were repeated independently two times.

### Contractile properties of TA muscle using sciatic nerve stimulation
TA muscle force properties were analyzed by using in situ muscle contraction measurements and nerve stimulation. Briefly, the sciatic nerve was carefully isolated and TA injured muscle was dissected free with its blood supply intact. The foot and the tibia of the mice were fixed by two clamps, and the distal tendon of the TA muscle was attached to a force transducer and positioned parallel to the tibia. Data were collected and stored for analyses with Chart v4.2.3 (PowerLab 4/25 ADInstrument, PHYMEP France). Throughout the experimental procedure, the mice were kept on a heating pad to maintain normal body temperature, and the muscles were continuously perfused with Ringer solution. Stimulation electrodes were positioned at the level of the sciatic nerve, and connected to a pulse generator with stimulation characteristics of 0.2 ms duration, 6 V stimulation amplitude. The muscle was stretched before stimulation voltage was applied to produce the most powerful twitch contractions. After a 3-min pause, twitch (0.2 ms, 6 V) et tetanic (200-ms burst, 6 V, 20 Hz) forces were recorded. At the end of the experiments, TA muscles were rapidly dissected and weighted. Twitch and tetanic forces were normalized in grams per milligram of fresh TA muscles.

### In vivo angiogenesis assay
Liquid Matrigel (10–16 mg of protein/ml) was dissolved at 4 °C in cold PBS at a final concentration equal to 1.0 μg/ml and mixed with FACS-sorted SCs from *Tg:Pax7nGFP* mice. Then, col Matrigel was injected subcutaneously (0.5 ml/mouse) with a cold syringe into the flank of C57BL/6 mice. The cold Matrigel that solidifies permits the penetration of host cells and the formation of new blood vessels. Six weeks after injection, mice were sacrificed and plugs were harvested, weighted and embedded in Tissue Tec OCT (Sigma-Aldrich), snap-frozen by immersion in liquid nitrogen-cooled isopentane and analyzed by immunofluorescence or stained for H&E. Images were obtained on Leica® TCS SPE DM 2500 microscope and LAS AF software (Leica®, Germany). For Matrigel plug, blood vessel formation into the plug was quantified in one section per plug, by counting the number of cells per mm2.

### Histological analysis
All histological analyses were performed in double blind and in the TA unless specified and in double blind. The TA was carefully dissected and snap frozen in liquid-nitrogen-cooled isopentane for a few minutes and stored at −80 °C prior to cryo-sectioning (10 μm sections). Sections were kept at room temperature overnight before staining. Sections were then rehydrated in PBS for 10 minutes and fixed in 10% formalin for 3 minutes. The sections were then routinely stained with hematoxylin and eosin (HE) using an automated stain machine or with Sirius Red to assess the collagen deposit. Sections were then analyzed using an automated Axioscan (Zeiss). For calcium deposits, their number was quantified on HE-stained TA sections and expressed as a number per section. For collagen deposit quantification, 10 different images randomly taken per section and 3 sections minimum per experimental group have been quantified using ImageJ software. We converted the pictures in a binary image and then collected the pixel

values. The collagen deposit area was quantified on TA section stained by Sirius Red among the total section area.

## Immunostainings

Immunostaining was performed on cryosections fixed with 4% paraformaldehyde (PFA EMS#15710) in cold PBS, permeabilized with 0.5% Triton X-100 20 min at room temperature, washed, and blocked with 10% BSA (Sigma-Aldrich, #A1933) for 30 min. Sections were incubated with primary antibodies overnight at 4 °C (Supplementary Table 1) and with Alexa-conjugated secondary antibodies 1/250 and Hoechst for 45 minutes. For BrdU immunostaining the cells were treated with 2 N HCl after fixation for 20 minutes at room temperature and neutralized with 0.1 M borate through 2 washes. The procedure was then identical to other immunostainings. Sections were then analyzed using an automated Axioscan (Zeiss) or Axio Observer Z1 (Zeiss).

## Cell count, sorting, and culture

For satellite cell counting, only the *tibialis anterior* muscle was dissected as previously described in cold DMEM-Glutamax (Gibco #41965-039)[68]. TA Muscles were then chopped with small scissors and put in a 50 ml Falcon tube with collagenase 0.08% (Sigma-Aldrich, #C51385), trypsin 0.08% (Sigma-Aldrich, #G6452) and DNAse 1UI/mL (Roche, #04716728001) at 37 °C with gentle agitation. After 20 minutes, the supernatant was collected in 20% serum placed on ice, and the collagenase/trypsin solution was added to continue the digestion. Once muscle is completely digested, the solution was filtrated using 40 µm cell strainers and analyzed by FACS. Cells were labeled with propidium Iodide 10 µg/ml (Sigma-Aldrich#P4170) to exclude dead cells and displayed using the PE (Phycoerythrin, Red) channel on the FACS profile. FACS analysis was done using a FACSaria (Beckman). All analyses and quantitation were performed using Summit v4.3 software from DakoCytomation and FloJo software. The quantification of GFP+ cells by cytometry was performed on the entire TA muscle on two biological replicates per condition of each independent experiment.

For the number of BrdU+ cells, GFP+ cells were FACS-sorted and plated in a cytospin machine (Cytospin4; Thermofisher), spun at 500 g with low acceleration for 5 min. The slide was dried off and cells were fixed with 4%PFA. After immunostaining against BrdU, proliferating cells were quantified on two biological replicates per condition and were expressed as the percentage of BrdU+ -GFP+ among the SCs.

For satellite cell FACS-sorting, all the muscles of the hind limbs from *Tg:Pax7nGFP mice* were dissected and digested as previously described. Cells were labeled with propidium Iodide 10 µg/ml to exclude dead cells and displayed using the PE channel on the FACS profile. Cell sorting was done with Aria III (BD Biosciences) and BD FACSDIVA software (BD Biosciences). SCs were directly plated onto 96-well plate (PerkinElmer, #6005550) coated with Matrigel (150 ng/ml, BD Biosciences#354234) at an initial density of 6000 cells per well. SCs were cultured in 1:1 DMEM-Glutamax (Gibco, #41965-039):Ham's F-12-Glutamax (Gibco #31765035) containing 20% serum FBS (Biowest S1860) designated as the regular medium containing endogenous 5-HT and they were kept in an incubator (37 °C, 5% CO2). For some in vitro experiments, serum was extracted from fluoxetine or control treated C57Bl6 mice after 6 weeks by heart puncture followed by centrifugation at 1500 g for 15 min. The thus-obtained supernatant replaced FBS in the culture medium; the rest of the medium was unchanged. The drug used was Fluoxetine (Sigma-Aldrich, #F132) at 1 µM and GR127935 (potent and selective 5-HT1B/1D antagonist, Tocris #1477/) at 0.1 µM. Then, cells were washed with PBS, fixed with 4% PFA for 10 min, stained according to immunofluorescence protocols. Fourteen images per well were acquired using 10x/NA 0.3 air objective on an Opera Phenix™ High-Content Imaging System (Perkin Elmer, Germany). The images were transferred to the Columbus Conductor™ software (Perkin Elmer Technologies) for data analysis.

For the isolation of SCs from the *Pax7-Cre^ER(T2)::tetO1B* mice, we used the five-step pre-platting as previously described[69]. Briefly, all muscles of the hind limbs of mice were dissected. After enzymatic digestion as previously described, the solution was seeded onto gelatin-coated 6-well plates. After incubation, the supernatant containing non-adherent cells was transferred to a new gelatin-coated well. The incubation times for each of the 5 stages respectively were 1 h, 2 h, 18 h, 24 h and 24 h. In the 4th and 5th pre-platting steps, adherent cells were transferred to new gelatin-coated wells for 1 h. Next the supernatant containing non-adherent SCs was transferred to flasks coated with Matrigel®. They were cultured for 3 days in 1:1 DMEM-Glutamax:Ham's F-12-Glutamax containing 20% serum FBS and kept in an incubator (37 °C, 5% CO2). The cells were then collected in RIPA buffer to protein extraction.

C2C12 cells between the 2nd and the 10th passage were maintained in a monolayer in DMEM-Glutamax (Gibco #41965-039) supplemented with 10% FBS (Biowest S1860) and kept in an incubator (37 °C, 5% CO2). For the proliferation assay, the cells were plated in 96-well plates (PerkinElmer, #6005550) at an initial density of 2000 cells per well and cultured in the regular medium with 10% FBS for 24 h. Then serotonin starvation was induced for 24 h by washing cells with PBS and replacing medium with 90% DMEM-Glutamax and 10% charcoal stripped FBS (Thermo, #A3382101). At 48 h post plating, C2C12 cells were exposed to different drugs for 24 and 48 h with medium changed every day. The drugs used were Fluoxetine (Sigma-Aldrich, #F132) at 1 µM, Serotonin (Sigma-Aldrich, #H9523) at 1 µM except for the increasing doses and the association of Fluoxetine1 µM + serotonin 0.1 µM, SB216641 (selective 5-HT1B antagonist, Tocris #1242) at 0.1 µM, GR127935 (potent and selective 5-HT1B/1D antagonist, Tocris #1477/) at 0.1 µM, MDL100907 (potent and selective 5-HT2A antagonist, Tocris #4173) at 0.1 µM. All the drugs used were dissolved in the water. For antagonist drugs, cells were pre-treated with specific antagonists for 30 min and then exposed to 5-HT or FLX combined with the antagonists for 24 h (two biological replicates per condition). Then, cells were washed with PBS, fixed with 4% PFA for 10 min and stained with Hoescht for 30 min, and were imaged using the 10x/NA 0.3 air objective on an Opera Phenix™ High-Content Imaging System (Perkin Elmer, Germany). The experimental procedures were repeated independently three times.

## Live video microscopy

FACS-sorted SCs were plated overnight on a 24-well glass bottom plate (P24G-0- 10-F; MatTek) coated with matrigel and placed in an incubator in the previously described medium. The plate was then incubated at 37 °C, 5% CO2 (Zeiss, Pecon). A Zeiss Observer.Z1 connected with a LCI PlnN 10×/0.8 W phaseII objective and AxioCam camera piloted with AxioVision was used. Cells (two biological replicates per condition) were filmed for up to 5 days, and images were taken every 30 min with brightfield and phase filters and MozaiX 3×3 (Zeiss). Raw data were transformed and presented as a video.

## Preparation and immunostaining of single muscle fibers

Extensor digitorum Longus muscles were carefully dissected and digested with 0.1% collagenase (Sigma-Aldrich, #C0130) for 1 h at 37 °C. The muscles were then flushed to separate single fibers. Fibers were then transferred one by one into a dish using serum-coated Pasteur pipettes. Cells were cultured in medium contained 40% DMEM-Glutamax with 1X Penicillin-streptomycin (Gibco), 40% MCDB201 (Sigma, #M6770), and 20% FBS and serotonin was used at 1 µmol for 4 days. Then, single fibers were fixed with 2% PFA in PBS at room temperature and immunostained in a 100-ml volume in 2-ml Eppendorf tubes. Images were acquired using a spinning disk confocal microscope (Cell Voyager - CV1000, Yokogawa®, Japan) with a 10x/NA 0.9 UPLSAPO objective.

## cAMP detection and Proteome profiler array

As previously described, C2C12 cells were plated on a 6-well plate at an initial density of 200,000 cells per well and cultured in the regular medium with 10% FBS for 24 h. Then serotonin starvation was induced for 24 h by washing cells with PBS and replacing medium with 90% DMEM-Glutamax and 10% charcoal stripped FBS. At 48 h post plating, C2C12 cells were exposed to different drugs for 30 min. The drugs used were Fluoxetine at 1 μM, Serotonin at 1 μM, GR127935 at 0.1 μM. For antagonist drugs, cells were pre-treated with specific antagonists for 30 min and then exposed to 5-HT combined with the antagonists for 30 minutes. cAMP was quantified using a cAMP ELISA Detection Kit (GenScript, #L00460) and the relative levels of protein phosphorylation was determined using a human phospho-kinase array kit (R&D, #ARY003B) as per the manufacturer's instructions, respectively. The experimental procedures were repeated independently three times.

## 5-HT dosage

Sera were centrifuged at 12,000 G for 5 min at 4 °C and the supernatant was collected. Samples were diluted to 1/10 in an ascorbate solution and then analyzed for 5-HT dosage using the high-performance liquid chromatography technique based on an ESA Coulochem III detector. The liquid phase (pH = 3.0) consisted of a solution of 75 mM of $NaH_2PO_4$, 25 μm EDTA, 1.7 mM of octanesulfonic acid and triethylamine in acetonitrile water (100 μl / L).

## Image analysis

All fiber parameters of each muscle section were performed automatically using the high-content analysis, MuscleJ[70], in ImageJ software environment. For diameter, area and number of fibers, as well as number of centro-nucleated fibers, sections were immunostained with Laminin and Hoescht independently of the centro or peripheral location of the nucleus. The quantification of SCs and differentiating cells by immunofluorescence was expressed as number of cells per mm2, when done in vivo. The total number of cells was counted and divided by the size (in mm²) of the tissue. SCs were quantified on section either after GFP or Pax7 immunostaining (specified in figure legends) on the entire section. At least 7 sections per muscle were quantified at different levels of the muscle. For the distribution of Pax7, Ki67 or myogenin markers within myogenic cells in vivo, after quantification of positive cells for each marker over the entire muscle section and reported per mm², the percentage of Pax7 + /Ki67+ or Ki67- cells was calculated by dividing the number of Pax7 + /Ki67+ cells and the Pax7 + /Ki67- cells respectively by the total number of Pax7+ cells; the percentage of Pax7 + /myogenin- and Pax7-/myogenin+ cells was calculated by dividing the number of Pax7 + /myogenin- cells and Pax7-/myogenin+ cells respectively by the total number of myogenic cells (i.e. the sum of the numbers of Pax7 + /myogenin+ and Pax7-/myogenin+ cells). Analysis of this distribution was carried out on two biological replicates of a single experiment. The quantification of vessels by immunofluorescence was expressed as number of cells per mm² or per fiber, when done in vivo. Vessels were quantified on section either after GFP or CD31 immunostaining (specified in figure legends) on the entire section. The quantification of immune cells (i.e., F4/80 or Gr1) was expressed as a percentage of infiltration area relative to the entire tissue section. The fiber typing distribution was performed automatically on the entire section.

Morphological analysis of cultured cells (i.e., SCs or C2C12s) was performed using Columbus Conductor™ software (Perkin Elmer Technologies). Cells touching the edges of the image were excluded. Columbus image analysis modules were used for feature extraction by well to assess the number of nuclei (detected by Hoechst stain), their surface area, roundness, and to measure the intensity within the nucleus of the immunofluorescence signal depending on the antibody used (Supplementary Table 1). The plasma membrane was detected with CellMask green staining to assess the number of nuclei per cell using the appropriate integrated image analysis modules. The number of nuclei was divided by the average number of nuclei in the control wells and the normalized data was analyzed as the percentage of nuclei. The nuclear intensity of each immunofluorescence was measured to define a threshold of positivity for each signal and then the percentage of immunofluorescence positive nuclei was calculated from the total number of nuclei detected in each well. The fusion index was calculated as the ratio of the number of nuclei in multinucleated cells ( ≥ 2 nuclei per cell) to the total number of nuclei per well.

## Western blot

Tissue samples were lysed with cold Ripa buffer (Sigma-Aldrich, #R0278) in Lysing Matrix Z tubes (MP Biomedicals, #116979010) and processed at 4.0 m/sec for 30 s with the FastPrep-24 instrument (MP Biomedicals, #116004500). Cell lysates were prepared by washing with cold PBS followed by incubation on ice for 1 h RIPA buffer with 1X complete protease inhibitor cocktail (Roche, #11873580001). To remove particulates, tissue and cell lysates were centrifuged at 14,000 × g for 15 min at 4 °C and the supernatant was collected. The protein suspensions were quantified using a Pierce BCA Protein Assay kit (Thermo, #23227) and the 96-well plates were read on GloMax® Discover (Promega) at 570 nm. 25 μg of proteins per sample were separated by electrophoresis on 10% SDS (Sigma-Aldrich, #L3771)−12% Acrylamide (Sigma-Aldrich, #A9099) gel and then transferred to a nitrocellulose membrane (PerkinElmer, #NBA083G001EA). The membranes were blocked with 5% BSA (Sigma-Aldrich, #A1933) in TBS (PBS with 0.1% Tween 20X (Sigma-Aldrich, #P9416)) for 1 h and then incubated with primary antibodies (Supplementary Table 2) overnight at 4 °C. After washed with TBS, the membranes were incubated with horseradish peroxidase (HRP) conjugated antibodies (Supplementary Table 2) for 45 min. Bound antibody complexes were visualized by SuperSignal West Femto Maximum Sensitivity Substrate (Thermo scientific, #34095). The exposed membranes were imaged using Gbox imaging system from Syngene (Cambridge, UK). The levels of GAPDH housekeeping protein expression were used for the normalisation of each target protein level.

## RT-qPCR

Total RNA was isolated from SCs using the RNAeasy Micro kit (Qiagen). The total RNA was reverse transcribed using Superscript® III Reverse transcriptase (Invitrogen). Real-time quantitative PCR was performed using Power Sybr Green PCR Master Mix (Applied Biosystems) and the rate of dye incorporation was monitored using the StepOneTM Plus RealTime PCR system (Applied Biosystems). At least three biological replicates were used for each condition. Data were analysed by StepOne Plus RT PCR software v2.1 and Microsoft excel. *GAPDH* transcript levels were used for the normalisation of each target (=ΔCT). RT-qPCR CT values were analysed using the $2^{-(\Delta\Delta Ct)}$ method to calculate the fold expression. The primers used were listed Supplementary Table 3.

## Luminex® (multiplex immunoassay)

As previously described, serum was extracted from fluoxetine or control treated animals after 6 weeks by heart puncture followed by centrifugation at 1500 × g for 15 min (n = 9 per condition). Then, supernatant was processed for analysis of Luminex® multiple cytokine and chemokine (Bio-Plex Pro™ Mouse Cytokine Standard 23-Plex, Group I and Standard 9-Plex, Group II).

## Statistics

Statistical analysis was performed using GraphPad Prism software version 9 using appropriate tests after being assessed for non-normal distribution: the non-parametric two-tailed Mann-Whitney test for comparing two groups, the non-parametric Kruskal-Wallis test for

comparing more than two groups, the non-parametric two-tailed Wilcoxon test for comparing to a theorical value such as statistical analysis of fold change, or Fisher's two-sided contingency test for the distribution of the percentage of cells, and a minimum of 95% confidence interval for significance; $p$ values indicated on figures are < 0.05 (*), < 0.01(**), < 0.001 (***), < 0.0001 (****), ns: not significant. Unless specified, quantitative data were represented by median and interquartile range.

### Reporting summary
Further information on research design is available in the Nature Portfolio Reporting Summary linked to this article.

## Data availability
All data generated or analyzed during this study are included in this article and its supplementary information files. The uncropped gel or blot figures and original data underlying Figs. 1–4 and Supplementary Figs. 1–5 are provided in a source data file. Further information and requests for data, resources and reagents should be directed to and will be fulfilled by the corresponding authors upon request. Source data are provided with this paper.

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

## Acknowledgements

We thank Johan Bedel, Rémy Dailleux, Marie Nguele, Cédric Thépenier, Aurélie Trignol (Institut Pasteur, France), Sophie Hüe and Mathieu Surenaud (Hôpital Henri Mondor, France), Indira David and Denis David (Université Paris Saclay, France), and Jacques Callebert and Jean-Marie Launay (Hôpital Lariboisière, France) for their scientific and technical help. The authors also thank Francine Coté and Olivier Hermine (Institut Imagine, Paris, France) who kindly provided the TPH1⁻/⁻ mice, Katherine Nautiyal (Columbia University, New York, USA) who kindly provided the *tetO1B* mice and Barbara Gayraud-Morel (Institut Pasteur, Paris, France) who kindly provided the *Pax7-Cre^(ER(T2))* mice. The authors also thank the histology, PFC and PBI platforms of the Institut Pasteur for their technical supports. UTechS PBI/C2RT is part of the France BioImaging infrastructure supported by the French National Research Agency (ANR-10-INSB-04-01, "Investments for the future") and is supported by Conseil de la Region Ile-de-France (Domaine d'Intérêt Majeur DIM1HEALTH) and

Fondation Française pour la Recherche Médicale (Program Grands Equipements). This work was financially supported by AFM-Téléthon and by the Fondation des Gueules Cassées received by FCh.

## Author contributions

M.F., M.B., P.R. conceived, planned, and carried out the experiments, analyzed the data, wrote the manuscript with input from all authors. D.B., M.J.M., A.D., A.H., D.H., A.T., F.C., C.H., A.L., A.K., R.M. participated in conception and carried out the experiments. A.D., O.M. revised the manuscript. R.G., F.C.h. designed and supervised the research and revised the manuscript.

## Competing interests

R.G. has received compensation as a member of the scientific advisory board of Janssen, Lundbeck, Novartis, Roche, SOBI, Takeda. He has served as a consultant and/or a speaker for Astra Zeneca, Boehringer-Ingelheim, Pierre Fabre, Eli Lilly, Lundbeck, LVMH, MAPREG, Novartis, Otsuka, Pileje, SANOFI, Servier and received compensation, and he has received research support from Servier. He is a board member of the Regstem company. O.M. has received consultancy fees from Astra-Zeneca, Blueprint Medicines, Boehringer-Ingelheim, Bristol-Myers Squibb, Eli Lilly, Ipsen, Merck Sharpe & Dohme, Pfizer, Roche, Servier, and Vifor Pharma. O.M. is an employee and shareholder of Amgen since Feb 1st, 2022. FCh is a board member of the Regstem company. The remaining authors have nothing to disclose.
