## [Peer Review File · Nature Communications]

Serotonin reuptake inhibitors improve muscle stem cell function and muscle regeneration in male miceReviewer #1 (Remarks to the Author):

In this manuscript the authors show that prolonged administration of the serotonin reuptake inhibitor, fluoxetine (FLX) in mice, increases the number of muscle stem cells and intramuscular angiogenesis, leading to improved regeneration as well as reduced fibrosis and inflammation. These effects were not observed in mice devoid of peripheral serotonin and upon pharmacological or genetic inactivation of the 5-HT_{1B} subtype serotonin receptor

While the primary observation that FLX promotes muscle progenitor cell expansion is of great interest in principle, the study is descriptive and mostly focused on cell counts, and mechanistic basis of this mitogenic effect remains unclear, thereby reducing somehow the interest of the finding.

This reviewer recommends addressing the following points

- 1) Is this a cell autonomous effect or it is related to interactions with other FLX-responsive cell types – i.e. endothelial or vessel associated cells. This should be tested experimentally by alternative knocking down 5-HT_{1B} specifically in MuSCs vs endothelial cells, using appropriate Cre-lox mice models.
- 2) Although the detailed investigation of the signal transduction and gene expression profile downstream to serotonin receptor might belong to an independent next investigation, it would be important to link the biological effect reported here to specific signaling cascade(s) and gene expression patterns in MuSCs and endothelial cells
- 3) Immunofluorescence analysis of myogenin/Pax7 in the same muscle stem cells within single fibers is recommended
- 4) The authors observed the same effect in both cultured C2C12 myoblasts and MuSCs, suggesting that the activation of serotonin receptor promotes an equivalent effect (i.e. mitosis) in committed myoblasts as well as muscle stem cells. Analysis of the physiological role of serotonin receptor in these two different cellular states is recommended.
- 5) Is the mitogenic effect promoting asymmetric cell division in MuSC?
- 6) Is there any physiological action of endogenous 5-HT_{1B}? KO mouse model?

Reviewer #2 (Remarks to the Author):

This study reports that a chronic treatment with fluoxetine increases the proliferation of skeletal muscle progenitor cells and boosts angiogenesis in muscles. It then uses an injury model to evaluate whether this treatment can enhance muscle regeneration. Using cell culture assays and transgenic mice it also shows that this effect is dependent on serotonin signaling through the 5HT_{1b} receptor.

Given the broad, and ever increasing, use of selective serotonin reuptake inhibitors this study is both timely and potentially important. The results presented are intriguing and novel. The authors need, however, to address some important technical and conceptual points.

A major concern is that the study relies only on cellular data. Metabolic and exercise testing should be performed to determine whether the changes observed translate into functional changes. This is both important for the un-injured and regeneration studies.

Another issue with the regeneration experiments is that the mice pre-treated for 6 weeks with fluoxetine start with a higher number of progenitors and therefore are expected to be regenerating faster than untreated mice. It is unclear whether the regeneration itself is enhanced by the fluoxetine treatment or that the treated mice have such a head start that they appear to regenerate faster. Control groups of un-injured mice (treated and not) as well as mice for which the treatment was terminated at the time of injury should be analyzed.

The in vitro data with C2C12 cells are confusing. If the effect of fluoxetine is mediated by its

binding to a serotonin receptor, the additive effect observed with fluoxetine and serotonin treatment is hard to explain since myoblasts do not synthesize serotonin; blocking SERT activity should not lead to increase extracellular serotonin levels as it happens in vivo. These experiments should be repeated with primary cell-sorted cells from the Tg:Pax7nGFP mice.

Additional points:

- The title should specify that the study was performed in mice.
- In Table 1, the SD for some important biomarkers such as MCP1, IL-6, TNF α , VEGF is very high. Given the low number of mice analyzed (n=4) this may have biased the statistics. Additional animals need to be analyzed to validate these results.
- Likewise, some of the panels shown in several other Figures (2, supp.2, 3G, supp 4, sup 5) present data with large SD and low number of mice. Additional animals should be analyzed to strengthen the data.
- Serotonin is not a stable molecule. The statement that FBS comprises endogenous 5-HT should be sustained by experimental data.
- As stated above, the effect shown with fluoxetine alone in sup. Figure 3A does not fit with a SERT/serotonin-mediated action of fluoxetine. These data should be experimentally strengthened, for example by using primary cells.
- In all experiments using GR127935 or other compounds, the data for these compounds alone should be shown.

Replies to the Reviewers' comments:
Please find the answers and corrections in blue below.

Reviewer #1 (Remarks to the Author):

In this manuscript the authors show that prolonged administration of the serotonin reuptake inhibitor, fluoxetine (FLX) in mice, increases the number of muscle stem cells and intramuscular angiogenesis, leading to improved regeneration as well as reduced fibrosis and inflammation. These effects were not observed in mice devoid of peripheral serotonin and upon pharmacological or genetic inactivation of the 5-HT_{1B} subtype serotonin receptor.

While the primary observation that FLX promotes muscle progenitor cell expansion is of great interest in principle, the study is descriptive and mostly focused on cell counts, and mechanistic basis of this mitogenic effect remains unclear, thereby reducing somehow the interest of the finding. This reviewer recommends addressing the following points.

We thank the reviewer for her/his positive comments on our work and constructive criticism of the manuscript.

1) Is this a cell autonomous effect or it is related to interactions with other FLX-responsive cell types – i.e. endothelial or vessel associated cells. This should be tested experimentally by alternative knocking down 5-HT_{1B} specifically in MuSCs vs endothelial cells, using appropriate Cre-lox mice models.

Our current results strongly support a cell autonomous effect of fluoxetine (FLX) on satellite cells (SCs) via the action of 5-HT/5-HT_{1B} receptor.

First, we showed that skeletal muscle and specifically primary SCs and immortalized myoblasts (C2C12) exhibited serotonergic identity through the expression of key players of the 5-HT system (**Fig.3a-b, Supplementary Fig.3b,3j**).

Furthermore, we previously demonstrated *in vitro* that FLX, in a 5-HT-dependent manner and upon 5-HT_{1B} receptor activation, promoted C2C12 cell division (**Supplementary Fig.3c-f**).

To unambiguously confirm the cell autonomous effects of FLX on SCs, we complemented our *in vitro* experimental approach by characterizing the effects of FLX on primary SCs sorted by FACS from *Tg:Pax7nGFP* mice. Because FBS serum is essential for culturing primary SCs, all these experiments included culture medium containing a concentration of endogenous 5-HT, which we measured by HPLC (**Supplementary Fig.3a**).

Thus, we showed that FLX exposure promoted SCs proliferation with a significant increase in the number of SCs at 2-, 4- and 14-days *post* plating (**Fig.3C**). Specifically, at 4 days *post* plating, FLX-exposed cells expressed increased numbers of Pax7+/GFP+ cells, a marker of early stages of myogenesis, as well as increased numbers of myogenin+ cells, a marker of early stages of myogenic differentiation (**Fig.3d-e**). Finally, at 14 days *post* plating, FLX-exposed cells showed a significant increase in the fusion index, a marker of the terminal stage of differentiation, associated with an increased number of single cells expressing Pax7+/GFP+, corresponding to the reserve pool of SCs (**Fig.3f-g**). The overall cellular effects induced by FLX were counteracted in the presence of a 5-HT_{1B} antagonist (**Fig.3c-g**).

In total, FLX exerts autonomous effects by targeting primary SCs that express the 5-HT_{1B} receptor and whose activation promotes the different stages of myogenesis with their proliferation, differentiation and self-renewal.

As suggested by the reviewer, we previously confirmed these results *in vivo* using an appropriate Cre-Lox model allowing specific deletion of 5-HT_{1B} receptor expression in SCs (**Supplementary Fig.4i**). Indeed, the effects of FLX targeting SCs were abolished in the basal state or *post* injury, after suppression of specific 5-HT_{1B} receptor expression by SCs, suggesting a vessel-independent effect of FLX (**Fig.4g, Supplementary Fig.4j**). Similarly, the positive effects of FLX on *post* injury collagen

deposition, one of the markers of effective wound healing, were also abolished in the absence of 5-HT_{1B} receptor expression by SCs (**Fig.4h**).

Associated with the specific effects of FLX on SCs, we showed that there were also other targets of this molecule such as vessels or immune system actors (**Fig.1d-g, Supplementary Fig.1h-j,2d-f**) that actively participate in the beneficial effects of FLX during muscle regeneration. These results are consistent with the literature since the effects of 5-HT/FLX on vascular tissue and the immune system have already been widely described in other organs (see References 9-10, 12-13, 56-63). More specifically, the positive effects of FLX during muscle regeneration were abolished when a 5-HT_{1B} antagonist was delivered systemically to the mice, highlighting the crucial role of this receptor regardless of the actors or their interactions.

The question of the cellular interaction between closed partners of muscle regeneration, such as vessels, macrophages, and other immune cells, but also fibro-adipogenic progenitors, pericytes etc., is quite relevant and will be the subject of a new study. Indeed, the novelty of the present work is carried by the demonstration of an autonomous effect of FLX on SCs.

2) Although the detailed investigation of the signal transduction and gene expression profile downstream to serotonin receptor might belong to an independent next investigation, it would be important to link the biological effect reported here to specific signaling cascade(s) and gene expression patterns in MuSCs and endothelial cells

As suggested by the reviewer, we performed different approaches on primary SC and C2C12 in order to identify signaling pathways underlying the biological effects of FLX/5-HT on SCs.

By RT-qPCR, we showed that FACS-sorted SCs from FLX-treated mice overexpressed genes related to myogenesis such as *Pax7* and *Myogenin*, as well as those related to cell division such as *Cyclin D1* (**Supplementary Fig.1g**).

Then, we investigated the intracellular levels of cAMP, as a previously known second messenger of the 5-HT₁ receptor family. After a short exposure to 5-HT, we showed a clear decrease in the intracellular level of cAMP in C2C12 cultured in a 5-HT-free medium and this effect was abolished with 5-HT_{1B} antagonist (**Supplementary Fig.3g**). Similarly, a short exposure to FLX in a regular medium resulted in a decrease in intracellular cAMP levels of C2C12 (**Supplementary Fig.3h**).

Finally, using a screening on phosphorylation changes of major protein kinases, we showed that a short exposure to 5-HT positively or negatively modulated, in a 5-HT_{1B} receptor-dependent manner, the activity of certain kinases (**Supplementary Fig.3i**).

We propose as a mechanistic hypothesis underlying the cellular effects of FLX that it, by blocking the SERT transporter, increases the bioavailability of extracellular 5-HT allowing the reinforcement of the signal induced by the activation of the 5-HT_{1B} receptor leading to the decrease of intracellular levels of cAMP, following the inhibition of adenylate cyclase, and to the positive or negative modulation of different key factors in the function of SCs such as ERK2 and Cyclin D1, which are well known to be involved in the proliferation process.

Furthermore, signaling pathways related to 5-HT_{1B} receptor activation and involved in angiogenesis have already been described in the literature (References 58-59). Specifically, the intracellular mediators of these effects are the production of NO and the activation of the Src/PI3K/Akt/mTOR/ER pathway.

Further investigations would clarify all the transduction pathways underlying the effects of serotonin, whether dependent on the 5-HT_{1B} receptor, on muscle stem cell function.

3) Immunofluorescence analysis of myogenin/Pax7 in the same muscle stem cells within single fibers is recommended

In our initial study, we analyzed Pax7 and Myogenin immunostaining during muscle regeneration and more precisely 4 days *post injury* (**Fig.2b-d**). At these early times of muscle regeneration, the muscle being too damaged, the isolation of single fibers is compromised.

However, as suggested by the reviewer, using isolated fibers from uninjured muscles, we studied *ex vivo* the differentiation status of SCs exposed for 4 days to 5-HT as shown in **Fig.3i**. We showed that 5-HT exposure promoted myogenic differentiation with an increase in the percentage of myogenin+ cells associated with a decrease in the percentage of Pax7+/GFP+ cells (**Fig.3h**).

Furthermore, as presented above, we confirmed *in vitro* the effects of FLX on early stages of myogenesis with an increased level of Pax7+/GFP+ and Myogenin+ cells for SCs exposed to FLX (**Fig.3d-e**).

4) The authors observed the same effect in both cultured C2C12 myoblasts and MuSCs, suggesting that the activation of serotonin receptor promotes an equivalent effect (i.e. mitosis) in committed myoblasts as well as muscle stem cells. Analysis of the physiological role of serotonin receptor in these two different cellular states is recommended.

We have previously shown *in vitro* that exposure to 5-HT and FLX, in a 5-HT-dependent manner, was associated with an increase in the rate of C2C12 cell division, suggesting a pro-proliferative effect of 5-HT/FLX (**Supplementary Fig.3c-d**). These effects were abolished in the presence of 2 different 5-HT1B antagonists without any effect of 5-HT2A antagonist as a control, suggesting a crucial role of the 5-HT1B receptor in the proliferative effects of 5-HT/FLX (**Supplementary Fig.3e-f**). In addition, 5-HT and 5-HT1B agonists also promoted terminal differentiation of myogenesis marked by an increase in the fusion index (**former Figure S3D**).

It should be first noted that the overall effects induced by FLX in a 5-HT/5-HT1B receptor-dependent manner, are not crucial for the physiological behavior of SCs as evidenced by the lack of abnormalities in muscle regeneration of basal SCs pool in *TPH1^{-/-}* or *Pax7-Cre^{ER(T2)}::tetO1B* mice compared to wild-type mice (**Fig.4a-b,g, Supplementary Fig.4a,j**).

Even if 5-HT/5-HT1B receptor stimulation is not mandatory for muscle, a pro-proliferative effect on SCs would be of major interest to improve muscle function in normal or pathological conditions.

As suggested by the reviewer and as mentioned previously, we deeply investigated the effects of FLX on FACS-sorted primary SCs cultures of 7 *Tg:Pax7nGFP* mice in medium with FBS containing endogenous 5-HT.

Thus, we showed that FLX exposure promoted SCs proliferation with a significant increase in the number of SCs at 2-, 4- and 14-days *post* plating (**Fig.3C**). Specifically, at 4 days *post* plating, FLX-exposed cells expressed increased numbers of Pax7+/GFP+ cells, a marker of early stages of myogenesis, as well as increased numbers of myogenin+ cells, a marker of early stages of myogenic differentiation (**Fig3d-e**). Finally, at 14 days *post* plating, FLX-exposed cells showed a significant increase in the fusion index, a marker of the terminal stage of differentiation, associated with an increased number of single cells expressing Pax7+/GFP+, corresponding to the reserve pool of SCs (**Fig.3f-g**). The overall cellular effects induced by FLX were counteracted in the presence of a 5-HT1B antagonist (**Fig.3c-g**).

In total, FLX exerts autonomous effects by targeting primary SCs that express the 5-HT1B receptor and whose activation promotes the different stages of myogenesis with their proliferation, differentiation and self-renewal.

As requested by the reviewer, we confirmed these results by another *in vitro* experimental approach with the culture of SCs in a medium comprising serum from FLX-treated and control mice instead of the usual FBS serum. In these sera, we previously characterized serotonin levels by HPLC that were increased in FLX-treated mice compared to controls (**Figure 1**, below). By Luminex, we also showed that FLX-treated mice had increased circulating levels of the growth factor FGFb compared to controls (**Table 1**). Although we did not have the opportunity to measure the plasma level of FLX in these sera, as this molecule has a long half-life ranging from 1 to 4 days¹, we can assume that serum from FLX-treated mice also contained the molecule.

Figure 1: 5-HT levels in sera from FLX-treated and control mice, measured by HPLC (n=3 mice per condition)

Thus, we showed by videomicroscopy that SCs cultured in serum from FLX-treated mice initiated their entry into the first cell division more rapidly and had a significantly higher rate of cell division than SCs cultured in serum from control mice (**Fig.3j-k**). Interestingly, the cell division promoting effects of FLX on SCs were abolished in the presence of a 5-HT1B antagonist (**Fig.3j-k**). In addition, SCs exposed to serum from FLX-treated mice for 4 days expressed an increased number of Myogenin+ cells and this FLX-induced myogenic differentiation effect was abolished by a 5-HT1B antagonist (**Fig.3l**). At 14 days *post* plating, exposure of SCs to serum from FLX-treated mice resulted in an increase in the number of Pax7+ cells, a marker of the reserve SC pool, and this effect was counteracted by a 5-HT1B antagonist (**Fig.3m**).

In total, the cellular effects induced by exposure to serum from FLX-treated mice were thus similar to those induced by exposure to fluoxetine alone and promoted, in a stimulation-dependent manner of the 5-HT1B receptor, activation with cell cycle entry, proliferation, early differentiation, and, finally, self-renewal of SCs. *In vivo*, we also showed that the beneficial effects of FLX during muscle regeneration included increased SCd proliferation and differentiation and were dependent on 5-HT and 5-HT1B receptor as demonstrated using pharmacological and genetic models (**Fig.4a-b,d-e,g**).

5) Is the mitogenic effect promoting asymmetric cell division in MuSC?

Indeed, our current results do not specify whether FLX through 5-HT has an exclusively proliferative effect targeting myoblasts by promoting symmetric cell division, which leads to an increase in cell density, notably *in vitro*, secondarily stimulating differentiation or whether these molecules show a dual effect stimulating both asymmetric and symmetric cell divisions promoting proliferation and differentiation, respectively.

Our *ex vivo* experimental approach analyzing the distribution of Pax7/GFP+ relative to Myogenin+ cells can be considered as an indirect estimation of the cell division state. Indeed, although the regulation of the cell division state remains complex and partially unknown, it has been described in SCs that the symmetric division state could result in identical daughter cells (both Pax7+ or Myogenin+, respectively), whereas the asymmetric division state could result in one Pax7+ and one Myogenin+ daughter cell ².

Thus, we previously found that exposure to 5-HT for 4 days resulted in an increase in the distribution of Myogenin+ cells associated with a decrease in Pax7+/GFP+ cells among SCs from single fibers, suggesting a 5-HT effect promoting asymmetric division (**Fig.3h-i**).

This question would naturally deserve further investigation in a future study. Moreover, this issue has already been studied in other stem cell cascades. For example, it was shown that 5-HT increased the rate of symmetrical division of neural progenitors without effect on neural stem cells and increased the proliferation and differentiation of myeloid progenitors without effect on hematopoietic stem cells (References 5, 36).

6) Is there any physiological action of endogenous 5-HT1B? KO mouse model?

In our study, using the conditional 5-HT1B receptor deletion mouse model specifically within SCs, we showed no abnormality of muscle regeneration, including SC behavior or collagen deposition, in tamoxifen-treated *Pax7-Cre^{ER(T2)}::tetO1B* mice compared to wild-type mice (**Fig.4g-h, Supplementary Fig.4j**).

To our knowledge, no studies have characterized the physiological muscle phenotype or during muscle regeneration of constitutive knockout of 5-HT1B receptor animal models. Furthermore, in studies using 5-HT1B KO mice, no pathological muscle phenotype was described, whereas other abnormalities were reported such as weight gain, increased bone formation, increased aggressive behavior, increased locomotor activity, decreased anxiety-related behavior, early age-related motor decline etc ³⁻⁸.

Reviewer #2 (Remarks to the Author):

This study reports that a chronic treatment with fluoxetine increases the proliferation of skeletal muscle progenitor cells and boosts angiogenesis in muscles. It then uses an injury model to evaluate whether this treatment can enhance muscle regeneration. Using cell culture assays and transgenic mice it also shows that this effect is dependent on serotonin signaling through the 5HT1b receptor.

Given the broad, and ever increasing, use of selective serotonin reuptake inhibitors this study is both timely and potentially important. The results presented are intriguing and novel. The authors need, however, to address some important technical and conceptual points.

We thank the reviewer for her/his positive comments on our work and for constructive comments in order to improve the manuscript.

A major concern is that the study relies only on cellular data. Metabolic and exercise testing should be performed to determine whether the changes observed translate into functional changes. This is both important for the un-injured and regeneration studies.

As suggested by the reviewer, we performed a functional study on muscle strength and physical performance parameters in FLX-treated mice in uninjured and injured muscle conditions.

Thus, we showed by an *in vivo* grip test that uninjured FLX-treated mice treated exhibited an increase in forelimb muscle strength (**Fig.1h**). In a treadmill exercise test, uninjured mice treated with FLX showed an improvement in performance with an increase in maximal aerobic velocity, distance traveled and exercise endurance (**Fig.1i, Supplementary Fig.1k-l**).

To further investigate the changes in muscle physiology induced by FLX treatment, we assessed muscle fiber typing. Indeed, FLX-treated mice showed a change in fiber typing on uninjured TA sections with a significant decrease in the percentage of fiber type IIX associated with a significant increase in the percentage of fiber I, IIA, IIB (**Fig.1j, Supplementary Fig.1m**).

In order to investigate the effect of FLX on muscle physiology during regeneration, a functional study on the *in situ* contractile parameters of the injured TA muscle was performed, using nerve-muscle stimulation to induce twitch and tetanos mechanical responses. At 14 days *post* notexine, we showed that the TA muscles from FLX-treated mice showed 22% and 27% increased relative amplitudes of twitch and tetanos contractile responses, respectively (**Fig.2i-j**).

Taken together, we have shown that FLX positively modifies muscle physiology with an increase in muscle strength and physical performance during exercise. These improved functional parameters are supported by changes in muscle metabolism with a change in the typing of muscle fibers, being enriched in fibers I, IIA, IIB corresponding to the fibers known as slow twitch with high oxidative activity, to the fibers known as fast twitch with high oxidative activity and to the fibers known as fast twitch with high glycolytic activity, respectively ⁹. In addition, we showed that the *in situ* contractile parameters of injured muscle were improved after treatment with FLX. These results are consistent with a recent study conducted by Tutakhail's team which also demonstrated an improvement in muscle performance after FLX treatment in mice (reference 66).

Another issue with the regeneration experiments is that the mice pre-treated for 6 weeks with fluoxetine start with a higher number of progenitors and therefore are expected to be regenerating faster than untreated mice. It is unclear whether the regeneration itself is enhanced by the fluoxetine treatment or that the treated mice have such a head start that they appear to regenerate faster. Control groups of

un-injured mice (treated and not) as well as mice for which the treatment was terminated at the time of injury should be analyzed.

As requested by the reviewer, we performed histological analysis at different time points of muscle regeneration in mice pre-treated with FLX at 18mg/kg/day for 6 weeks, mice treated with FLX only *post* notexine (NTX) injury and control mice (**Figure 2a**).

Figure 2: Muscle regeneration is harmoniously improved only by fluoxetine delivered by a 6-

week pre-treatment before injury

(a) Schematic representation of the notexine (NTX)-induced TA injury model, fluoxetine (FLX) administration and death time points. (b) Quantification of SCs number by Pax7 immunostaining 4 days *post* injury on TA sections from C57Bl6 mice treated with control, 6w-FLX pre NTX, FLX *post* NTX (n=7-14 mice per condition). (c) Quantification of differentiating cells number by Myogenin immunostaining 4 days *post* injury on TA sections from C57Bl6 mice treated with control, 6w-FLX pre NTX, FLX *post* NTX (n=7-14 mice per condition). (d) Representative HE-stained TA sections of C57Bl6 mice treated with control, 6w-FLX pre NTX, FLX *post* NTX at 14 days *post* NTX injury. Scale bars indicate 100 μ m. (e) Automatic quantification of the centro-nucleated fibers number 14 days *post* injury on TA sections from C57Bl6 mice treated with control, 6w-FLX pre NTX, FLX *post* NTX using the MuscleJ analysis (n=7-14 mice per condition). (f) Automatic quantification of the fibers number 14 days *post* injury on TA sections from C57Bl6 mice treated with control, 6w-FLX pre NTX, FLX *post* NTX using the MuscleJ analysis (n=7-14 mice per condition). (g) Percentage of F4/80+ (macrophages) and (h) Gr1+ (granulocytes) immune cells infiltration areas 14 days *post* injury on TA sections from C57Bl6 mice treated with control, 6w-FLX pre NTX, FLX *post* NTX and analyzed with Fiji software (n=7-14 mice per condition). (i) Quantification of the calcium deposit number by Hematoxylin and Eosin staining 14 days *post* injury on TA sections from C57Bl6 mice treated with control, 6w-FLX pre NTX, FLX *post* NTX using the MuscleJ analysis (n=7-14 mice per condition). (j) Percentage of the collagen deposit area stained with Sirius Red 14 days *post* injury on TA sections from C57Bl6 mice treated with control, 6w-FLX pre NTX, FLX *post* NTX and analysed with Fiji software (n=7-14 mice per condition). All values are represented as median with interquartile range. * p \leq 0.05, ** p \leq 0.01, ***p \leq 0.001, ****p \leq 0.0001.

At 4 days *post* injury, as expected, we observed that mice pre-treated with FLX showed an increase in SCs as well as Myogenin + differentiating cells (**Figures 2b-c**). Mice treated with FLX *post* injury showed an equal increase in the number of SCs but had a similar number of Myogenin+ cells compared to the control mice (**Figures 2b-c**).

At 14 days *post* injury, both FLX-pretreated and FLX-post-NTX-treated mice had increased numbers of muscle fibers and centro-nucleated fibers (**Figures 2d-e**), as well as decreased immune cell infiltration (**Figures 2f-g**). However, in contrast to the FLX-pretreated mice, the mice treated by FLX *post* NTX had similar calcium and collagen deposition as the control group (**Figures 2h-j**).

As previously shown, pre-treatment with FLX for 6 weeks prior to muscle injury supports a harmonious acceleration of muscle regeneration through stimulating muscle stem cells and myogenesis, modulating inflammation and resulting in efficient wound healing marked by reduced collagen deposition.

When FLX is delivered only *post* injury, it also exerts early positive effects targeting the pool of activated SCs and myoblasts. These results are consistent with the cell-autonomous effects of FLX demonstrated *in vivo* in the uninjured muscle (**Supplementary Fig.1e-f**) and *in vitro* via faster entry into cell division, increased rate of cell division resulting in increased proliferation of SCs (**Fig.3c-d,j-k**). Thus, these results suggest that FLX can rapidly induce a cell-autonomous effect stimulating the proliferation of already activated SCs *post* injury, unlike quiescent SCs in uninjured muscle, requiring prolonged exposure to fluoxetine (**Supplementary Fig.1e-f**).

In contrast, a short *post* injury administration of FLX does not result in an effect targeting the early stages of myogenic differentiation, in contrast to our previous *in vivo* and *in vitro* results (**Fig.2d, Fig.3e,l**). Interestingly, despite the lack of effect promoting early myogenic differentiation, FLX delivered *post* injury also results in an increased number of fibers including regenerated ones, suggesting a similar acceleration of muscle regeneration as in case of 6 weeks FLX pretreatment. However, the muscle regeneration induced by FLX delivered *post* injury is less harmonious and efficient as shown by the similar or even increased number of collagen and calcium deposits compared to control mice, although the infiltration of immune cells is simultaneously decreased.

Taken together, these results suggest that fluoxetine requires prolonged pre-injury administration to any muscle injury, suggesting indispensable upstream tissue and cellular remodeling to support its full triple beneficial action on muscle regeneration. Interestingly, fluoxetine delivered post-injury may induce beneficial effects on muscle regeneration, specifically targeting SCs early.

The *in vitro* data with C2C12 cells are confusing. If the effect of fluoxetine is mediated by its binding to a serotonin receptor, the additive effect observed with fluoxetine and serotonin treatment is hard to

explain since myoblasts do not synthesize serotonin; blocking SERT activity should not lead to increase extracellular serotonin levels as it happens *in vivo*. These experiments should be repeated with primary cell-sorted cells from the Tg:Pax7nGFP mice.

As suggested by the reviewer, we have clarified the mechanisms of action underlying the *in vitro* effects of FLX on SCs and C2C12. Indeed, our mechanistic hypothesis is that FLX, by blocking the SERT transporter known to be its main target of action, increases the bioavailability of extracellular 5-HT, leading to a prolonged stimulation of the 5-HT1B receptor, mediator of the cellular effects of FLX.

First, we showed that skeletal muscle and specifically primary SCs as well as immortalized myoblasts (C2C12) exhibited serotonergic identity through the expression of key players of the 5-HT system, including the SERT transporter, the main target of SSRIs (**Fig.3a-b, Supplementary Fig.3b,3j**).

Next, we previously showed *in vitro* that 5-HT exerted a proliferative effect on C2C12s marked by an increase in division rate in a dose-dependent manner (**Supplementary Fig.3d**). Interestingly, when C2C12s were exposed to FLX in a medium devoid of endogenous 5-HT, no effect on the rate of cell division was observed, unlike C2C12s exposed to FLX + exogenous 5-HT, which resulted in an increase in the rate of cell division (**Supplementary Fig.3c**). This proliferative effect was also observed when C2C12s were exposed to FLX in a culture medium comprising FBS with endogenous 5-HT (**Supplementary Fig.3a,f**). These results thus highlight that 5-HT is crucial in mediating the cellular effects of FLX on C2C12s. They are further consistent with *in vivo* results showing abolished effects of FLX on SCs during muscle regeneration in TPH1^{-/-} mice lacking peripheral 5-HT (**Fig.4a-b**).

As suggested by the reviewer and as mentioned previously, we deeply investigated the effects of FLX on FACS-sorted primary SCs cultures of 7 Tg:Pax7nGFP mice in medium with FBS containing endogenous 5-HT.

Thus, we showed that FLX exposure promoted SCs proliferation with a significant increase in the number of SCs at 2-, 4- and 14-days *post* plating (**Fig.3C**). Specifically, at 4 days *post* plating, FLX-exposed cells expressed increased numbers of Pax7+/GFP+ cells, a marker of early stages of myogenesis, as well as increased numbers of myogenin+ cells, a marker of early stages of myogenic differentiation (**Fig3d-e**). Finally, at 14 days *post* plating, FLX-exposed cells showed a significant increase in the fusion index, a marker of the terminal stage of differentiation, associated with an increased number of single cells expressing Pax7+/GFP+, corresponding to the reserve pool of SCs (**Fig.3f-g**). The overall cellular effects induced by FLX were counteracted in the presence of a 5-HT1B antagonist (**Fig.3c-g**).

As requested by the reviewer, we confirmed these results by another *in vitro* experimental approach with the culture of SCs in a medium comprising serum from FLX-treated and control mice instead of the usual FBS serum. Thus, we showed by videomicroscopy that SCs cultured in serum from FLX-treated mice initiated their entry into the first cell division more rapidly and had a significantly higher rate of cell division than SCs cultured in serum from control mice (**Fig.3j-k**). Interestingly, the cell division promoting effects of FLX on SCs were abolished in the presence of a 5-HT1B antagonist (**Fig.3j-k**). In addition, SCs exposed to serum from FLX-treated mice for 4 days expressed an increased number of Myogenin+ cells and this FLX-induced myogenic differentiation effect was abolished by a 5-HT1B antagonist (**Fig.3l**). At 14 days *post* plating, exposure of SCs to serum from FLX-treated mice resulted in an increase in the number of Pax7+ cells, a marker of the reserve SC pool, and this effect was counteracted by a 5-HT1B antagonist (**Fig.3m**).

Taken together, FLX exerts autonomous effects by targeting primary SCs that express the SERT transporter and the 5-HT1B receptor by promoting different stages of myogenesis, including their activation, proliferation, differentiation and self-renewal, in a 5-HT and 5-HT1B receptor dependent manner.

Additional points:

- The title should specify that the study was performed in mice.

As suggested by the reviewer, we have corrected the title to " Serotonin reuptake inhibitors improve muscle stem cell function and muscle regeneration in mice ".

- In Table 1, the SD for some important biomarkers such as MCP1, IL-6, TNF α , VEGF is very high. Given the low number of mice analyzed (n=4) this may have biased the statistics. Additional animals needs to be analyzed to validate these results.

As requested by the reviewer, we wished to replicate the Luminex experiment using TA muscle homogenates under the different conditions presented. Unfortunately, due to the high sensitivity of the Luminex technique between the different batches of kits, some of our results were not consistent with the first series, despite our increase of the n value per condition. Thus, due to the lack of robustness of these results, we removed them from our study.

However, we demonstrated by Luminex on a sufficient n value (9 animals per condition) with low SD, the modifications induced by a prolonged treatment with FLX on the serum levels of cytokines, chemokines and growth factors. Thus, we showed that FLX-treated mice had increased circulating levels of the growth factor FGF compared to controls (**Table 1**).

- Likewise, some of the panels shown in several other Figures (2, supp.2, 3G, supp 4, sup 5) present data with large SD and low number of mice. Additional animals should be analyzed to strengthen the data.

As suggested by the reviewer, we increased the number of animals for the muscle regeneration experiments induced by intramuscular injection of notexine (**Fig. 2b,d,g-h, Supplementary Fig.2a,d-f**). Of note, because the results are from nonparametric statistical analysis, they are represented by median with interquartile range and not by mean with standard deviation.

For Figure 3G, we have strengthened and deepened our characterization of the effects of fluoxetine on MuSCs *in vitro* and replaced with the new **Fig.3c-g**, representing n=7 mice/ condition.

The experiments in **Supplementary Fig.4c-h** and **Supplementary Fig.5a-g** could be replicated and completed with the antagonist control condition alone.

- Serotonin is not a stable molecule. The statement that FBS comprises endogenous 5-HT should be sustained by experimental data.

As suggested by the reviewer, we measured by HPLC endogenous 5-HT concentration in FBS serum (**Supplementary Fig.3a**).

- As stated above, the effect shown with fluoxetine alone in sup. Figure 3A does not fit with a SERT/serotonin-mediated action of fluoxetine. These data should be experimentally strengthened, for example by using primary cells.

As suggested by the reviewer and as mentioned previously, we have clarified the mechanisms of action underlying the *in vitro* effects of FLX and we deeply investigated the effects of FLX on primary SCs (**Fig. 3c-g,j-m**).

- In all experiments using GR127935 or other compounds, the data for these compounds alone should be shown.

Except for the *in vitro* experiments on C2C12s, all *in vitro* and *in vivo* experiments with 5-HT1B and 5-HT2A antagonists were completed with the antagonist alone condition (**Fig. 3c-g, Supplementary Fig.4c-h and Supplementary Fig.5a-g**).

REFERENCES:

1. Hiemke, C. & Härtter, S. Pharmacokinetics of selective serotonin reuptake inhibitors. *Pharmacology and Therapeutics* vol. 85 11–28 Preprint at [https://doi.org/10.1016/S0163-7258\(99\)00048-0](https://doi.org/10.1016/S0163-7258(99)00048-0) (2000).
2. Tierney, M. T. & Sacco, A. Satellite Cell Heterogeneity in Skeletal Muscle Homeostasis. *Trends in Cell Biology* **26**, 434–444 (2016).
3. Saudou, F. *et al.* Enhanced aggressive behavior in mice lacking 5-HT1B receptor. *Science (1979)* **265**, 1875–1878 (1994).
4. López-Rubalcava, C., Hen, R. & Cruz, S. L. Anxiolytic-like actions of toluene in the burying behavior and plus-maze tests: Differences in sensitivity between 5-HT(1B) knockout and wild-type mice. *Behavioural Brain Research* **115**, 85–94 (2000).
5. Buhot, M. C. *et al.* Protective effect of 5-HT1B receptor gene deletion on the age-related decline in spatial learning abilities in mice. *Behavioural Brain Research* **142**, 135–142 (2003).
6. Sibille, E. *et al.* Lack of serotonin1B receptor expression leads to age-related motor dysfunction, early onset of brain molecular aging and reduced longevity. *Molecular Psychiatry* **12**, 1042–1056 (2007).
7. Nautiyal, K. M. *et al.* Distinct Circuits Underlie the Effects of 5-HT1B Receptors on Aggression and Impulsivity. *Neuron* **86**, 813–827 (2015).
8. Nautiyal, K. M. *et al.* A lack of serotonin 1B autoreceptors results in decreased anxiety and depression-related behaviors. *Neuropsychopharmacology* **41**, 2941–2950 (2016).
9. Schiaffino, S. & Reggiani, C. Fiber types in mammalian skeletal muscles. *Physiol Rev* **91**, 1447–1531 (2011).

Reviewer #1 (Remarks to the Author):

The authors have addressed some of the reviewer's comments, but important concerns remain on the mechanism by which FLX promotes MuSC activity. In particular, it remains unclear whether FLX breaks quiescence, induces cell cycle progression or differentiation, as the data seem to suggest. By RT-qPCR, the showed that FACS-sorted SCs from FLX-treated mice overexpressed genes related to myogenesis such as Pax7 and Myogenin, as well as those related to cell division such as Cyclin D1(Supplementary Fig.1g). Pax7, cyclinD1 and Myogenin are mutually exclusive markers of 3 different stages of MuSC – quiescence, proliferation and differentiation, respectively. As RT-qPCR is an assay from bulk cells, it is important to determine the relative level of expression of these three markers by immunofluorescence in order to define the response to FLX by relative proportions of MuSCs. This seems consistent with multiple signalling pathways elicited by FLX in MuSCs that however have not been delineated by the authors. It is important that the authors further clarify this issue in order to conclusively established how FLX promotes MuSC activity.

Reviewer #2 (Remarks to the Author):

The authors have addressed my concerns

Replies to the Reviewers' comments:

Please find the answers and corrections in blue below.

Reviewer #1 (Remarks to the Author):

The authors have addressed some of the reviewer's comments, but important concerns remain on the mechanism by which FLX promotes MuSC activity. In particular, it remains unclear whether FLX breaks quiescence, induces cell cycle progression or differentiation, as the data seem to suggest. By RT-qPCR, the showed that FACS-sorted SCs from FLX-treated mice overexpressed genes related to myogenesis such as Pax7 and Myogenin, as well as those related to cell division such as Cyclin D1(Supplementary Fig.1g). Pax7, cyclinD1 and Myogenin are mutually exclusive markers of 3 different stages of MuSC – quiescence, proliferation and differentiation, respectively. As RT-qPCR is an assay from bulk cells, it is important to determine the relative level of expression of these three markers by immunofluorescence in order to define the response to FLX by relative proportions of MuSCs. This seems consistent with multiple signalling pathways elicited by FLX in MuSCs that however have not been delineated by the authors. It is important that the authors further clarify this issue in order to conclusively established how FLX promotes MuSC activity.

We thank the reviewer for her/his positive comments on our work and constructive criticism of the manuscript.

As suggested by the reviewer, we have clarified the results of our study regarding the mechanisms of action of FLX, with particular attention to the cellular behavior of SCs.

Indeed, we have shown *in vivo* that FLX promotes the emergence from quiescence and the proliferation of SCs, allowing an almost doubling of the SCs pool in the absence of muscle injury (**Figures 1b-c and Supplementary Figures 1b, c, f, g**). Furthermore, after 6 weeks of FLX treatment, 90% of SCs had divided at least once (**Supplementary Figure 1f**) and 4% were still dividing at week 6 (**Supplementary Figure 1g**). This result was consistent with the increase in *Cyclin D1* gene expression observed in FACS-sorted SCs from uninjured muscle after 6 weeks of FLX treatment (**Supplementary Figure 1h**). Because the SCs population is highly heterogeneous, it is not surprising to observe distinct effects of FLX on SCs activity, including quiescence, proliferation, and/or differentiation. In fact, we have shown that in the absence of muscle injury, 10% of the SCs pool appeared unaffected by FLX after 6 weeks of treatment, highlighting a varied pattern of SCs response to FLX (**Supplementary Figure 1f**).

In order to clarify our results and as suggested by the reviewer, we have corrected the **Supplementary Figure 1h**. Regarding the relative expression of the different markers in the SCs population at steady-state, the FACS-sorted cells used in RT-qPCR were isolated from *Tg:Pax7nGFP* mice and thus all expressed the transcription factor PAX7 (**Supplementary Figure 1a**). Although BrdU and Cyclin D1 are not the same markers, they are both still markers of cell division. Thus, we can estimate that when SCs expressed a higher level of the *Cyclin D1* gene at 6 weeks of FLX treatment, 4% of these SCs are still dividing (**Supplementary Figure 1g**). Finally, we attempted to demonstrate myogenin immunostaining on muscle sections, but in the absence of muscle injury, we did not observe myogenin positive cells (**data not shown**). This result suggests that the effects of FLX on gene expression of myogenesis genes such as *myogenin* are not always consistent with their protein expression. Because the results are confusing, we corrected this figure (**Supplementary Figure 1h**).

Complementary to the cell-autonomous effects of FLX demonstrated *in vivo*, we showed *in vitro* that FLX exerted direct effects on SCs in a 5-HT/5-HT1B receptor axis-dependent manner by promoting:

- Cell activation with a faster entry into the cell cycle, equivalent to a faster exit from quiescence (**Figure 3j**)
- cell proliferation with an increased rate of cell division (**Figures 3c-d,k**). These results are consistent, for example, with the increased activity of the Akt and TOR signaling pathways by 5-HT demonstrated in C2C12s (**Supplementary Figure 3i**).
- an enhancement of early and terminal differentiation (**Figures 3e-f,h-i,l**). These results are also consistent with the increased activity of ERK2 and STAT5b signaling pathways by 5-HT demonstrated in C2C12s (**Supplementary Figure 3i**).
- A maintained self-renewal of SCs without depletion of the SCs pool (**Figures 3g,m**).

These results could be further consolidated, and as the reviewer rightly points out, important mechanistic questions remain. However, the precise mechanisms of action of serotonin on stem cells are complex, as evidenced by the vast literature on the effects of serotonin on adult hippocampal neurogenesis which have not hitherto been elucidated ¹. It has been demonstrated the expression of at least 10 5-HT receptors by neural stem cells and progenitors, notably cell type 1, 2a and 2b, and which can modulate multiple signaling pathways involved in this neurogenesis ^{1,2}. This complexity of the 5-HT system and its modes of action are also found in other organs, such as angiogenesis (reference 57).

Given the known complexity of the mechanisms of action of 5-HT, it seems relevant to continue our efforts to investigate the mechanistic and cellular effects of 5-HT/FLX on SCs but in a future study specifically targeting this issue.

Indeed, the present study has brought to light a major and unexpected discovery on the positive effects of FLX on muscle regeneration associated with a functional and metabolic improvement of striated muscle. It deserves to be shared with the broad readership of *Nature Communication*, encompassing scientists from different backgrounds, and it opens major perspectives of translational medicine with the investigation in clinical practice of the therapeutic potential of FLX in muscle diseases.

Reviewer #2 (Remarks to the Author):

The authors have addressed my concerns

We thank the reviewer for her/his previous comments on our work.

References:

1. Alenina, N. & Klempin, F. *The role of serotonin in adult hippocampal neurogenesis. Behavioural Brain Research* vol. 277 49–57 (2015).
2. Wirth, A., Holst, K. & Ponimaskin, E. How serotonin receptors regulate morphogenic signalling in neurons. *Progress in Neurobiology* vol. 151 35–56 Preprint at <https://doi.org/10.1016/j.pneurobio.2016.03.007> (2017).

Reviewer #1 (Remarks to the Author):

Unfortunately the authors have not satisfactorily addressed my concerns on the mechanism by which FLX promotes MuSC activity. They tried to provide a verbal explanation, but did not perform the requested analysis of Pax7, cyclinD1 and Myogenin expression in MuSCs within the 3 different stages – quiescence, proliferation and differentiation - in response to FLX. This analysis is absolutely feasible and can be done in vivo (on tissue sections) as well as ex vivo (by single fiber analysis) in wild type mice at different time points after injury (typically days 1, 3 and 5 post-injury) in untreated or FLX treated mice. Specific antibodies and experimental conditions are available from previous studies published by multiple investigators in the field.

Replies to the Reviewers' comments:

Please find the answers and corrections in blue below.

Reviewer #1 (Remarks to the Author):

Unfortunately the authors have not satisfactorily addressed my concerns on the mechanism by which FLX promotes MuSC activity. They tried to provide a verbal explanation, but did not perform the requested analysis of Pax7, cyclinD1 and Myogenin expression in MuSCs within the 3 different stages – quiescence, proliferation and differentiation - in response to FLX. This analysis is absolutely feasible and can be done *in vivo* (on tissue sections) as well as *ex vivo* (by single fiber analysis) in wild type mice at different time points after injury (typically days 1, 3 and 5 post-injury) in untreated or FLX treated mice. Specific antibodies and experimental conditions are available from previous studies published by multiple investigators in the field.

We thank the reviewer for her/his positive comments on our work and constructive criticism of the manuscript.

As suggested by the reviewer, we performed an immunofluorescence-based kinetic analysis over time of cell distribution *in vivo* of the expression of the markers Pax7, Ki67 (another well-known cell division marker) and Myogenin within the MuSC population on muscle tissue sections from FLX-treated and untreated C57Bl6 mice before injury and throughout muscle regeneration after injury.

In the absence of muscle injury, mice treated with FLX for 6 weeks showed an increase in the population of Pax7-positive/Ki67-positive MuSCs compared with control mice (**Supplemental Figure 1h and Figure 1, below**), while the percentage of Pax7-negative/Myogenin-positive MuSCs was low and similar in both groups (**Supplemental Figure 1i and Figure 1, below**).

These results are consistent with our previous experiments showing that FLX led to an increase in MuSCs number by promoting cell division and upregulation of *Cyclin D1* gene expression, allowing an almost doubling of the MuSCs pool in the absence of muscle injury (**Figures 1b-c and Supplementary Figures 1b, c, f, g**).

Taken together, these results suggest that in the absence of muscle injury, FLX promotes the emergence from quiescence and the proliferation of MuSCs without affecting their differentiation capacity.

The kinetics over time of cell distribution of MuSCs' markers Pax7, Ki67 and Myogenin have previously been finely characterized during muscle regeneration^{1,2}. In the early stages of muscle regeneration, it was shown that injury initially induced a drastic reduction in the MuSC pool and, two days after injury, most MuSCs remained in a quiescent state and a minority were activated without dividing. Five days after injury, MuSCs were dividing strongly, and the differentiation capacity of the myogenic cell population increased. The switch from quiescence to cell proliferation and differentiation of MuSCs occurred between the second day and the fifth *post* injury. Thus, we studied MuSC activity at four days *post* injury, enabling us to appreciate at one time point consistent with our previous experiments, the different stages of the MuSC cascade: emergence from quiescence, proliferation, and differentiation.

At four days *post* injury, FLX-treated mice showed an increase in both levels of Pax7-positive/Ki67-positive MuSCs and Pax7-negative/Myogenin-positive MuSCs compared with control mice (**Supplemental Figures 1a-b**).

These results are consistent with our previous experiments showing that FLX-treated mice showed increased numbers of Pax7 and Myogenin-positive MuSCs number four days *post* injury (**Figures 2b-e**).

Taken together, these results suggest that FLX promotes the proliferation and differentiation of MuSCs during the early phases of muscle regeneration.

The cell-autonomous effects of FLX demonstrated *in vivo* on MuSC activity are also consistent with the *in vitro* positive effects of FLX that we previously described on MuSCs, namely stimulation of cell activation with accelerated entry into the cell division cycle, stimulation of cell proliferation with an increased rate of cell division, and stimulation of differentiation capacities (**Figures 3c-e, h-k**).

Beyond 7 days *post* injury, it has previously been shown that the proliferation and differentiation capacities of MuSCs declined, suggesting that the myofiber repair process had been completed, giving

way to the muscle tissue remodeling phase¹. We therefore pursued the kinetic analysis of cell distribution of our markers Pax7, Ki67 and Myogenin within MuSCs at a late stage of muscle regeneration (i.e. 14 days *post injury*) to assess the evolution of MuSCs activity.

At fourteen days *post injury*, the rates of Pax7-positive/Ki67-positive and Pax7-negative/Myogenin-positive MuSCs were similar in both groups and were lower than those at early times of muscle regeneration (**Figure 1, below**).

These results suggest that during muscle regeneration, on the one hand, FLX exerts a stimulatory action on MuSC activity at an early stage and, on the other hand, the action of FLX on MuSC activity is controlled, allowing a return to usually reduced MuSC activity at a late stage. These data are consistent with our previous results showing that FLX led to an acceleration of muscle regeneration, which remained harmonious (**Figures 2f-h and Supplemental Figures 2e-k**).

Thus, as suggested by the reviewer, we have clarified the results of our study regarding the mechanisms by which FLX promotes MuSCs activity.

Figure 1: Kinetics over time of cell distribution of Pax7/Ki67 immunostaining (top) and Pax7/Myogenin immunostaining (bottom) among SCs on TA sections from control and FLX-treated (6 weeks of treatment) C57Bl6 mice before injury, four days and fourteen days *post* injury (n=5-7 mice per condition).

References:

1. Borok, M. *et al.* Progressive and Coordinated Mobilization of the Skeletal Muscle Niche throughout Tissue Repair Revealed by Single-Cell Proteomic Analysis. *Cells* **10**, (2021).
2. Hardy, D. *et al.* Comparative Study of Injury Models for Studying Muscle Regeneration in Mice. *PLoS One* **11**, e0147198 (2016).

Reviewer #1 (Remarks to the Author):

The authors have responded to the request to perform an immunofluorescence-based kinetic analysis over time of cell distribution in vivo of Pax7, Ki67 and Myogenin within the MuSC population on muscle tissue sections from FLX-treated and untreated C57Bl6 mice before injury and throughout muscle regeneration after injury, by providing only bar graphs. However, it is fair that the authors also provide representative pictures of these immunofluorescence analysis, as it is important from a reviewer standing point to look at the quality & resolution of the images before they are quantified and reported in graphs. It is also important that the authors clearly state the number of experimental replicates and number of cells analyzed.

Replies to the Reviewers' comments:

Please find the answers and corrections in blue below.

Reviewer #1 (Remarks to the Author):

The authors have responded to the request to perform an immunofluorescence-based kinetic analysis over time of cell distribution in vivo of Pax7, Ki67 and Myogenin within the MuSC population on muscle tissue sections from FLX-treated and untreated C57Bl6 mice before injury and throughout muscle regeneration after injury, by providing only bar graphs. However, it is fair that the authors also provide representative pictures of these immunofluorescence analysis, as it is important from a reviewer standing point to look at the quality & resolution of the images before they are quantified and reported in graphs. It is also important that the authors clearly state the number of experimental replicates and number of cells analyzed.

We thank the reviewer for her/his positive comments on our work and constructive criticism of the manuscript.

As suggested by the reviewer, we have clarified our previous experiment, which characterized the kinetics over time of the cell distribution of the MuSCs' markers Pax7, Ki67 and myogenin in the steady state and during the early stages of muscle regeneration.

First, we provided representative immunostainings of each marker on muscle section (**Supplemental Figures 2b and 2d; Figure 1 below**). Next, we clarified the analysis methods used in the image analysis section of the methods section. Finally, we added all the raw data (i.e., number of cells per mm² and percentage of positive cells of interest) to the source data file.

Thus, as suggested by the reviewer, we have clarified the method used for the results of our study regarding one of the mechanisms by which FLX promotes MuSCs activity.

Figure 1: Representative immunostainings of Pax7, Ki67 and myogenin markers among SCs on TA sections from control and FLX-treated (6 weeks treatment) C57Bl6 mice four days *post injury*. Sections (a) and (b) show merged and separate illustrations of the markers Pax7 (green), Ki67 (red), Hoechst (nuclei, blue). Sections (c) and (d) show merged and separated illustrations of the markers Pax7 (green), myogenin (red), Hoechst (nuclei, blue). Scale bars indicate 50 μ m.

Reviewer #1 (Remarks to the Author):

The authors have provided IF images of sub-optimal quality and resolution, showing random fields of muscle sections without key details (i.e. laminin staining) that are typically required for identification of muscle stem cells - i.e., sub-laminar position.

It also remains unclear how many replicates have been generated for each experimental point throughout the whole manuscript.

Since most of conclusions of this manuscript are based on IF analysis, this reviewer remains highly concerned.

Replies to the Reviewers' comments:

Please find the answers and corrections in blue below.

Reviewer #1 (Remarks to the Author):

The authors have provided IF images of sub-optimal quality and resolution, showing random fields of muscle sections without key details (i.e. laminin staining) that are typically required for identification of muscle stem cells - i.e., sub-laminar position.

It also remains unclear how many replicates have been generated for each experimental point throughout the whole manuscript.

Since most of conclusions of this manuscript are based on IF analysis, this reviewer remains highly concerned.

We thank the reviewer for her/his positive comments on our work and constructive criticism of the manuscript.

As requested by the reviewer, we have added the low-resolution whole sections of muscle tissue and improved the quality of representative illustrations of Pax7, Ki67 and myogenin immunostaining at four days *post* injury (**Figures 1a-b and 2a-b below**).

At four days *post* injury induced by notexin, as we are in the early phase of muscle regeneration, we can observe major necrosis of muscle fibers, infiltration of immune cells, the onset of vascular remodeling, and activation of the pool of muscle stem cells that are able to migrate out of the sublaminar localization where they are usually found in the quiescent state¹⁻³. In this early phase of muscle regeneration, the quality of immunostaining is usually reduced by the high background induced by tissue necrosis. Thus, Laminin labeling is also impacting by the destruction of muscle tissue architecture, and it is not mandatory for the identification of muscle stem cells, that are otherwise specifically labeled by the transcription factor Pax7. Therefore, we did not perform Laminin labeling in our other immunostaining experiments performed at four days *post* injury, that already showed high quality of Pax7 and myogenin specific immunolabeling (**Figures 2c-f**). This approach is also consistent with previous seminal studies that have characterized muscle stem cells during the early phases of muscle regeneration after injury in mice^{1,2}. **Figure 3 below**, taken from the above-mentioned seminal articles, shows the degradation of muscle tissue architecture four days after muscle injury induced by notexin.

As requested by the reviewer, we have also clarified our method (in the methods section, in the legend of each figure and in the raw data source file) by specifying the number of biological replicates and the number of independent experiments performed for each figure. For immunofluorescence quantification of the distribution of markers Pax7, Ki67, myogenin within muscle stem cells, it was already specified that this was carried out on two biological replicates from the same experiment.

Taken together, we have answered the question of the effects of fluoxetine *in vivo* on the distribution of Pax7, Ki67, myogenin markers within muscle stem cells before and after muscle injury, added representative images for each type of immunofluorescence used in our work and finally clarified our method of biological replicates and independent experiment replicates throughout the manuscript. Thus, we hope to have satisfied the remaining concerns of the reviewer.

Figure 1: Representative immunostainings of Pax7 and Ki67 markers among SCs on TA sections from control (a) and FLX (b) -treated (6 weeks treatment) C57Bl6 mice four days *post injury*. Sections (whole section and area enlargement delimited by red square) show merged and separate illustrations of the markers Pax7 (green), Ki67 (red), Hoechst (nuclei, blue), the arrows indicate the Pax7+/Ki67+ SCs. Scale bars indicate 50 μ m.

Figure 2: Representative immunostainings of Pax7 and myogenin markers among SCs on TA sections from control (a) and FLX (b) -treated (6 weeks treatment) C57Bl/6 mice four days *post injury*. Sections (whole section and area enlargement delimited by red square) show merged and separated illustrations of the markers Pax7 (green), myogenin (red), Hoechst (nuclei, blue). Scale bars indicate 50 μ m.

Figure 3: Representative illustrations of muscle histology at different time points after notexin injury, taken from seminal articles on muscle injury models in mice. (a) Hematoxylin and eosin on a muscle section at 18h, 2 days, 4 days, 12 days, and one month after notexin injury, taken from Hardy et al.² (b) Hematoxylin and eosin on a muscle section at 12h, 2 days, 4 days, 7 days, and 10 days after notexin injury, taken from Arnold et al.¹

References:

1. Arnold, L. *et al.* Inflammatory monocytes recruited after skeletal muscle injury switch into antiinflammatory macrophages to support myogenesis. *J Exp Med* **204**, 1057–1069 (2007).
2. Hardy, D. *et al.* Comparative Study of Injury Models for Studying Muscle Regeneration in Mice. *PLoS One* **11**, e0147198 (2016).
3. Gayraud-Morel, B., Chrétien, F. & Tajbakhsh, S. Skeletal muscle as a paradigm for regenerative biology and medicine. *Regenerative Med* **4**, 293–319 (2009).

Reviewer #1 (Remarks to the Author):

The authors have satisfactorily addressed my major concern and I have no further concerns regarding publishing their interesting findings.

Replies to the Reviewers' comments:
Please find the answers and corrections in blue below.

Reviewer #1 (Remarks to the Author):

The authors have satisfactorily addressed my major concern and I have no further concerns regarding publishing their interesting findings.

We thank the reviewer for her/his positive comments on our work and constructive criticism of the manuscript.